# Demystifying Language Model Forgetting with Low-rank Example Associations

**Xisen Jin, Xiang Ren**
University of Southern California
{xisenjin,xiangren}@usc.edu

## Abstract

Large language models (LLMs) suffer from forgetting of upstream knowledge when fine-tuned. Despite efforts on mitigating forgetting, few have investigated how forgotten upstream examples are dependent on newly learned tasks. Insights on such dependencies enable efficient and targeted mitigation of forgetting. In this paper, we empirically analyze forgetting that occurs in $N$ upstream examples of language modeling or instruction-tuning after fine-tuning LLMs on one of $M$ new tasks, visualized in $M \times N$ matrices. We show that the matrices are often well-approximated with low-rank matrices, indicating the dominance of simple associations between the learned tasks and forgotten upstream examples. Leveraging the analysis, we predict forgetting of upstream examples when fine-tuning LLMs on unseen tasks with matrix completion over the empirical associations. This enables fast identification of most forgotten examples without expensive inference on the entire upstream data. Despite simplicity, the approach outperforms prior approaches that learn semantic relationships of learned tasks and upstream examples with LMs. We demonstrate the practical utility of our analysis by showing statistically significantly reduced forgetting as we upweight predicted examples for replay during fine-tuning.

## 1   Introduction

There has been a growing need for continued fine-tuning of LLMs to mitigate harmful behaviors, update outdated knowledge, and adapt to unseen tasks and domains. Although fine-tuning allows efficient and incremental adaptation of models, models may suffer from catastrophic forgetting (McCloskey & Cohen, 1989; Goodfellow et al., 2014) of upstream knowledge learned in the pre-training or instruction-tuning phase, causing unintended prediction changes over known information. This is problematic for the performance and reliability of LLMs deployed online, limiting the feasibility of continual fine-tuning in practice (Raffel, 2023; Shi et al., 2024).

Existing works demonstrate that replaying or mixing in past examples is effective and scalable approaches to mitigate LLM forgetting (Scialom et al., 2022; Roth et al., 2024; Li et al., 2024b; Ibrahim et al., 2024; Ye et al., 2024). These approaches, however, often rely on random sampling of past examples; knowing what models forget after fine-tuning allows more efficient and targeted mitigation of forgetting – *e.g.*, by prioritizing the replay of more forgotten examples (Toneva et al., 2019; Aljundi et al., 2019a). In this paper, we explore how forgetting caused by unseen tasks can be efficiently predicted, and more specifically, from the forgetting that occurred while learning other tasks. The complexity of associations between learned tasks and forgotten examples plays an important role in predictability; Figure 1 (a) illustrates a hypothetical scenario where certain upstream examples suffer more forgetting regardless of the learned tasks, making forgetting easily predictable; in contrast, (b) exemplifies upstream example forgetting that is highly dependent on the learned tasks. Existing theoretical and empirical studies on the associations between learned and forgotten

39th Conference on Neural Information Processing Systems (NeurIPS 2025).

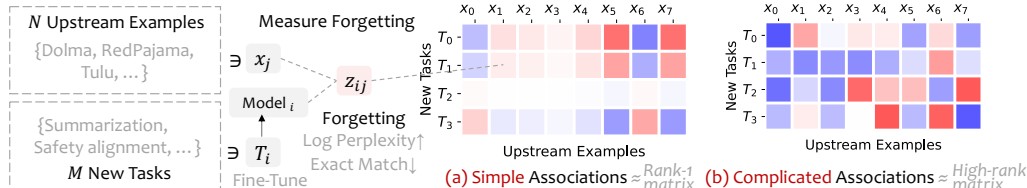

Figure 1: The problem setup of analyzing the associations between learned tasks and forgotten upstream examples as we fine-tune LLMs on one of unseen new tasks. Over total $N$ upstream examples and $M$ unseen tasks, we measure and record forgetting (in red) in a $M \times N$ matrix and attempt to fit the associations with low-rank approximations. Better low-rank approximations indicate simpler associations between learned tasks and forgotten upstream examples.

tasks focus on shallower models (Lee et al., 2021; Goldfarb et al., 2024; Ramasesh et al., 2021); the problem is under-explored for LLM forgetting or in an example-level. Swayamdipta et al. (2020); Maini et al. (2022) characterize training examples that are prone to forgetting, but they do not touch on how example forgetting depends on the learned tasks.

Specifically, we start by analyzing the associations between the learned tasks and forgotten upstream examples in LLM fine-tuning. We measure forgetting (in continuous log perplexity increase or binary exact match drop) over $N$ upstream examples, after fine-tuning the model on one of $M$ unseen instruction-tuning tasks, while summarizing the results in a $M \times N$ matrix. We evaluate the complexity of the associations by measuring the goodness-of-fit of low-rank approximations of the example associations. We then examine how the complexity of the example associations varies across model types (OLMo, OLMo2, MPT, Pythia) and sizes (1B to 13B parameters).

Our findings suggest that the associations between learned tasks and forgotten examples are often well-approximated with low-rank matrices. On OLMo-1B and OLMo-7B, rank-3 approximation fits the associations between 85 learned tasks and 140,000 upstream examples with $R^2 > 0.69$. We notice that the forgetting of more capable and recent LLM families are more complicated, requiring higher-rank approximations; within the same model family, the complexity of the associations remains stable or increases with the model size. The matrix decomposition further interprets the associations by distinguishing forgetting that are independent of or dependent on what the model learns.

Following the low-rank approximations of the associations, we predict example forgetting on unseen tasks by solving a matrix completion problem over the association matrices, analogous to collaborative filtering (Sarwar et al., 2001) in recommender systems, achieving both efficiency and interpretability. Our matrix factorization (MF) or $k$-nearest neighbor (KNN) models outperform previous approaches that learn semantic relations of two examples with LMs (Jin & Ren, 2024). As an example, we achieve 58.16 F1 in predicting binary example forgetting where the F1 of random guess is only 6.4.

Lastly, we demonstrate the practical benefit of predicting forgetting in mitigating forgetting. We upweight upstream examples with higher predicted forgetting during replay as we fine-tune LLMs on new instruction-tuning tasks, achieving statistically significant improvements in alleviating forgetting over held-out upstream examples compared to replaying random examples.

To summarize, the contributions of this paper are (1) an empirical analysis on how forgotten examples are associated with learned tasks in representative 1B to 13B language models, and (2) a novel approach of predicting example forgetting by solving a matrix completion problem over the empirical associations, and (3) a scalable and efficient algorithm to mitigate forgetting during LLM fine-tuning by upweighting upstream examples for replay according to the predicted forgetting.

## 2   Problem and Analysis Setup

In this section, we start by formally defining the metrics of forgetting and set up the problem formulation of analyzing the associations between learned tasks and forgotten upstream examples. We then introduce models and datasets that were used to collect the statistics.

## 2.1 Collecting Statistics of Forgetting

**Upstream examples and learned tasks.** LLMs are commonly pre-trained with language modeling objectives over massive collections of corpora, and optionally post-trained (instruction-tuned) to better follow human instructions. We use *upstream data* to refer to language modeling or instruction tuning training data used at the pre-training or post-training phase of LLMs. For upstream data of language modeling, we define each upstream example $x_j \in x_{1..N}$ as a chunk of a document (*e.g.,* a Wikipedia article) of a model-specific maximum number of tokens. For instruction tuning, each $x_j \in x_{1..N}$ corresponds to a pair of instructions and ground truth responses.

**Measuring forgetting.** We fine-tune an LLM (or an instruction-tuned LLM) on one unseen instruction-tuning task $T_i$ from a collection of tasks $T_{1..M}$. This results in $M$ separately fine-tuned models $f_{1..M}$. We then evaluate performance degradation on each upstream example $x_j \in x_{1..N}$. We use log perplexity as the main performance metric as it is applicable to both language modeling and instruction tuning, and is known to correlate well with other dataset-specific metrics (Hoffmann et al., 2022). For instruction tuning tasks with a restricted output space (*e.g.,* multi-choice questions), we also measure binary exact-match accuracy (EM). We measure forgetting $z_{ij}$ that occurs on an upstream example $x_j \in x_{1..N}$ as increase in log perplexity or drop in exact match after fine-tuning the LM on a new task $T_i \in T_{1..M}$. We record forgetting $z_{ij}$ in an association matrix $Z \in \mathbb{R}^{M \times N}$.

## 2.2 Models and Datasets

Our analysis requires open access to upstream data of LLMs. We perform main experiments with OLMo-1B, OLMo-7B, and OLMo-7B-Instruct (Groeneveld et al., 2024). We also include OLMo2-7B, OLMo2-13B (Walsh et al., 2024), MPT-7B (Databricks, 2023), and Pythia-1B to 12B (Biderman et al., 2023) for studying the complexity of the example associations across model types and sizes.

Table 1: Summary of experiment setups. We measure forgetting on upstream examples $x_{1..N}$ after fine-tuning the models over one of the new tasks $T_{1..M}$. We measure upstream example forgetting in either log perplexity increase or exact match drop.

| Model | Upstream $x_j$ | Learned Tasks $T_i$ | $|T_{1..M}|$ |
|---|---|---|---|
| OLMo | Dolma | FLAN, Tulu V2, Dolly | 85 |
| MPT | Redpajama | Tulu V2, Dolly | 19 |
| Pythia | Pile | Tulu V2, Dolly | 19 |
| OLMo2 | OLMo2-Mix | Tulu V2, Dolly | 19 |
| OLMo-Instruct | Tulu V2, FLAN | MMLU, BBH, TruthfulQA, Dolly, OLMo2-SFT-Mix | 142 |

**Upstream examples $x_{1..N}$ where forgetting is evaluated.** For OLMo, OLMo2, MPT, and the Pythia family, we evaluate log perplexity increase over Dolma (Soldaini et al., 2024), OLMo2-Mix (Walsh et al., 2024), Redpajama (TogetherAI, 2023), and Pile (Gao et al., 2020) respectively, each corresponding to their upstream pretraining corpora. We sample $10k$ to $140k$ documents truncated into 2,048 tokens. For OLMo-Instruct, we evaluate log perplexity increase on Tulu V2 (Ivison et al., 2023) which the model is instruction-tuned on. For the FLAN (Longpre et al., 2023) subset of Tulu, we also measure the drop of binary exact matches over correctly predicted upstream examples before fine-tuning.

**Unseen Task $T_{1..M}$ where models are fine-tuned.** We fine-tune non-instruction-tuned models over 66 tasks from FLAN, 11 tasks from Tulu and 8 tasks from Dolly (Conover et al., 2023). For OLMo-Instruct, we fine-tune OLMo-7B-Instruct over new task data from MMLU (Hendrycks et al., 2021), BBH (Suzgun et al., 2022), TruthfulQA (Lin et al., 2022), Dolly, and OLMo2-SFT-Mix (Walsh et al., 2024). Table 1 summarizes the setups. We fine-tune full model parameters with a 2e-6 learning rate and other consistent configurations. Training details are included in Appendix C.

## 3 Associations between Learned Tasks and Forgotten Examples

In this section, we analyze the associations between learned tasks and forgotten upstream examples represented in the $M \times N$ association matrices $Z$. We visualize the association matrix $Z$ collected from the setups described in Sec. 2. We formally define low-rank approximations and set up quantitative metrics of the complexity of the associations in Sec. 3.1, and examine the results of approximation across model types and sizes in Sec. 3.2. Lastly, we try to interpret the extracted associations in Sec. 3.3.

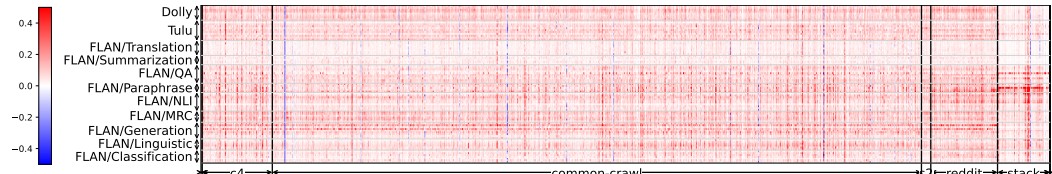

Figure 2: An example of visualized association matrix of forgetting $Z \in \mathbb{R}^{M \times N}$ between $M = 85$ learned tasks and $N = 141,876$ upstream examples (from Dolma) on OLMo-7B. Each pixel $z_{ij}$ indicates forgetting (in log-perplexity increase) that occurs on an upstream example $x_j$ (in $x$-axis) after fine-tuning the model on a task $T_i$ (in $y$-axis). We annotate the domains (*e.g.*, reddit) of upstream examples in the $x$-axis and the category of each learned task (*e.g.,* FLAN/QA) in the $y$-axis. We include visualizations of more models and setups in Figure 6 and Figure 8 in Appendix.

## 3.1 Methods and Metrics of Approximating Example Associations

To examine whether simple patterns are dominant in the example associations represented by $Z$, we attempt to approximate $Z$ with low rank matrices. When $Z$ represents the increase in log perplexity (also known as the loss of language modeling), we fit matrix factorization models $Z^r = \sum_{k=1}^r \boldsymbol{\alpha}_k \boldsymbol{\beta}_k^T$ that minimize the Frobenius norm $||Z - Z^r||_F$, where $\boldsymbol{\alpha}_k \in \mathbb{R}^M$, $\boldsymbol{\beta}_k \in \mathbb{R}^N$, $r$ is the rank of the matrix decomposition and $k$ is the index of the component. For binary forgetting measured with exact match drop, we fit a logistic matrix factorization model $Z^r = \sigma(\sum_{k=1}^r \boldsymbol{\alpha}_k \boldsymbol{\beta}_k^T)$ that minimizes the cross entropy between $Z^r$ and $Z$, where $\sigma$ is the sigmoid function. We measure $R^2$ or F1 scores of the approximation as the goodness-of-fit metrics.

**Interpretations.** When $r = 1$, the approximation effectively assumes that the forgetting $z_{ij}$ (or its logit) is a scalar product of a parameter $\beta^{(j)}$ specific to each upstream example and each newly learned task $\alpha^{(i)}$. The set of more forgotten upstream examples is independent of which task the model learns, as $\beta^{(j)}$ trivially determines how fast forgetting uniformly happens on all upstream examples. With a higher rank $r$, the approximation captures task-dependent forgetting where certain upstream examples are disproportionately more forgotten when learning certain tasks. This inner product formulation is also connected to the first-order approximation of loss increase as an inner product of weight updates and gradients (Lopez-Paz & Ranzato, 2017; Lee et al., 2019), in which case $r$ is the number of LLM parameters.

## 3.2 Examining Complexity of Example Associations

**General findings.** We present $R^2$ or F1 of fitting the association with $Z^r$ with progressively higher rank $r$ in Fig. 3(a). We notice that across all training setups of OLMo models, $R^2$ quickly increases to 0.69 with $r = 3$. Notably, even the rank-1 approximation $Z^1$ can achieve $R^2$ scores higher than 0.5. The results suggest that simple patterns are dominant in the associations between learned tasks and forgotten examples.

**Example associations across model types and sizes**. We compare $R^2$ of the approximations with a fixed $M$ and $N$ over Pythia, MPT, OLMo, and OLMo2 models. Fig. 3(b) and (c) summarize $R^2$ at a given rank $r$. We notice that (1) the goodness of approximations differs among model types. On Pythia and MPT, the $R^2$ scores at $Z^3$ are higher than 0.88, while on OLMo-7B and OLMo2, the $R^2$ scores are around 0.75 and 0.65. (2) The size of the models within the same model family also has an impact on $R^2$. On Pythia and OLMo2, $R^2$ stays stable with a slight decrease as the model size increases from 1B to 13B. On OLMo, $R^2$ is noticeably lower on 7B models compared to 1B. (3) Model families that forget more (*e.g.,* Pythia) tend to produce simpler example associations (higher $R^2$). However, within the same model family, OLMo-7B results in a lower $R^2$ score than OLMo-1B despite the fact that the average forgetting is higher.

To summarize, we empirically observe that the associations between learned tasks and forgotten upstream examples are more complicated in more recent and capable LLMs, requiring higher-rank approximations. The complexity of the associations stays stable or increases with larger models within the same family. In Appendix K, we provide more intuitions about how model capability and sizes affect the complexity of the associations with a set of synthetic experiments over MNIST and

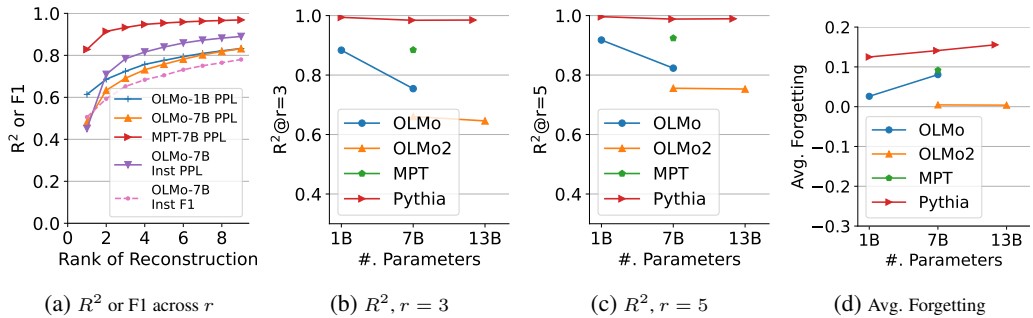

(a) $R^2$ or F1 across $r$     (b) $R^2$, $r = 3$     (c) $R^2$, $r = 5$     (d) Avg. Forgetting

Figure 3: (a) $R^2$ or F1 of the low-rank approximations as we progressively increase the rank of the reconstruction. Forgetting is measured with log perplexity increase or exact match (EM) drop. In (b) and (c), we compare $R^2$ of approximations at a given rank $r$ across models of different types and sizes over the fixed $M = 19$ tasks from Tulu and Dolly. We also report average upstream example forgetting in (d) as a reference, and include more statistics in Table 5 in Appendix.

Table 2: Correlations between various measures of similarity and the actual upstream example forgetting represented in the association matrix $Z$.

| Model | OLMo-1B | | OLMo-7B | |
|---|---|---|---|---|
| Metrics | Pearson $\rho$ | Spearman $\rho$ | Pearson $\rho$ | Spearman $\rho$ |
| *Token or representational similarity* | | | | |
| TF-IDF | -0.049 | -0.035 | 0.048 | 0.137 |
| OLMo 1B Final Layer | 0.021 | 0.017 | 0.059 | 0.062 |
| SBERT | 0.053 | 0.072 | 0.055 | 0.094 |
| OpenAI/text-embedding-3-small | 0.050 | 0.073 | 0.165 | 0.224 |
| *First order approximations* | | | | |
| $\langle$Gradient, Weight differences$\rangle$ | -0.003 | -0.009 | -0.004 | -0.004 |
| $\langle$Gradient, Gradient$\rangle$ | 0.061 | 0.052 | 0.013 | 0.002 |
| *Forgetting occurred when learning another task* | | | | |
| $\text{Avg}_{p \neq q}\, \rho(Z_{p,:}, Z_{q,:})$ | 0.582 | 0.410 | 0.400 | 0.367 |

multi-layer MLPs. In Appendix H, we further study the effect of model training configurations (*e.g.*, learning rate, batch size) on the complexity of the associations $Z$. We consider that the low-rank approximation is prevalent across setups.

### 3.3 Interpreting Example Associations

**Patterns of forgetting from matrix factorizations.** The matrix factorization of $Z$ yields interpretable patterns of forgetting in each of its components $\boldsymbol{\alpha}_k \boldsymbol{\beta}_k^T$. As an example, Fig. 9 in Appendix visualizes patterns captured by the $k$-th component in OLMo-7B experiments. We notice the patterns interesting, yet semantically intriguing. For example, on OLMo-7B, Stackoverflow documents are less forgotten when learning summarization tasks, while more forgotten when learning certain paraphrasing tasks.

**Correlations between example associations and similarity measures.** Are the associations between learned tasks and forgotten examples interpretable from the similarity between the learned tasks and upstream examples? We consider (1) textual similarity measures, such as token or representational similarity, and (2) first-order approximations, such as inner products of gradients and inner products between weight updates and gradients (Lee et al., 2019; Doan et al., 2020). For token similarity, we use TF-IDF vectorized representations. For representational similarity, we use representations given by OLMo-1B, SBERT (Reimers & Gurevych, 2019), or OpenAI text-embedding-3-small model. We detail each similarity measure in Appendix E. We then evaluate the correlations between the actual forgetting $z_{ij}$ and the various similarity measures and summarize the results in Table 2. For reference, we also include average correlations between each pair of rows of $Z$, *i.e.*, forgetting on upstream examples after learning two different tasks $T_p, T_q \in T_{1..M}$, defined as $\frac{1}{(M-1)^2} \sum_{p \neq q} \rho(Z_{p,:}, Z_{q,:})$.

As shown in Table 2, first-order approximations can achieve statistically significant correlation $|\rho| < 0.1$ with the actual forgetting, and textual similarities achieve $|\rho| < 0.25$ with the actual

forgetting. Although the results imply that similarity measures can signal forgetting, the correlations are far below the average correlations between any two rows in $Z$. Therefore, we hypothesize that leveraging the statistics of forgetting allows better prediction of forgetting than the contents of the tasks and examples, which elicits our next research question about predicting example forgetting.

# 4 Predicting Example Forgetting with Association Matrix Completion

We utilize our findings in Sec. 3 to predict example forgetting as the model learns a new task, a problem also studied in prior works (Jin & Ren, 2024), and effectively mitigate forgetting. We then perform targeted replay by prioritizing more forgotten examples. Although the ground truth forgetting can be directly obtained by running inference with the fine-tuned model over the upstream data, this incurs extensive computational cost; in contrast, once a prediction model is trained, forgetting caused by unseen tasks can be predicted efficiently.

Following the analysis in Sec. 3, we formulate prediction of example forgetting as a matrix completion problem over the empirical associations $Z$. We start by setting up the problem formulation of predicting example forgetting, and evaluate the performance of different matrix completion algorithms. We then demonstrate the practical benefit of predicting example forgetting by utilizing the prediction outcomes to mitigate forgetting during fine-tuning.

## 4.1 Training and Evaluation of Forgetting Prediction

Our goal is to accurately predict forgetting $z_{ij}$ over upstream examples $x_{1..M}$ when the model is fine-tuned on an unseen task $T_j$ with a prediction model $g$, without running expensive LLM inference on all $x_{1..M}$. To evaluate this, we create training and test splits by partitioning the set of fine-tuning tasks (noted as $\mathcal{T}_{\text{train}}$ and $\mathcal{T}_{\text{test}}$) and the rows of the association matrices $Z$. We further control whether $\mathcal{T}_{\text{train}}$ and $\mathcal{T}_{\text{test}}$ belong to the same category of tasks to test both in-domain and out-of-domain generalization ability of the prediction models. For OLMo-1B and 7B experiments, we use FLAN as in-domain tasks and Tulu and Dolly as out-of-domain testing tasks. For OLMo-7B-Instruct experiments, we use MMLU, BBH, OLMo2-SFT-Mix as in-domain tasks and use TruthfulQA and Dolly as out-of-domain testing tasks. Details about the tasks included in the training, in-domain testing, and out-of-domain testing sets are discussed in Tables 15 and 16 in Appendix D.

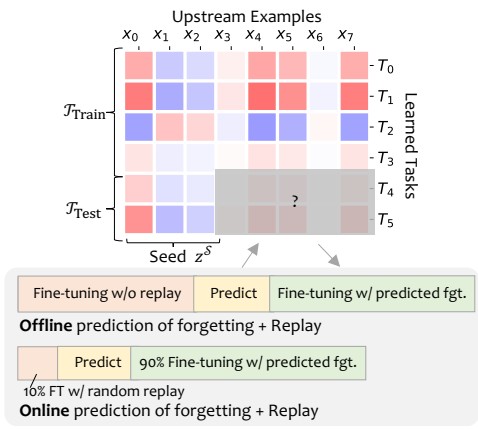

Figure 4: The training and testing setup of predicting example forgetting with association matrix completion, and their integration into example replay methods to mitigate forgetting.

To apply matrix completion for predicting forgetting, a few entries $z_{ij}$ should be known when a new fine-tuning task $T_i \in \mathcal{T}_{\text{test}}$ (row $i$) is introduced. We therefore assume access to the ground truth forgetting $z_{ij}$ of a tiny random set $\mathcal{S}$ ($|\mathcal{S}| = 30$) of upstream examples for $T_i \in \mathcal{T}_{\text{test}}$, noted as seed forgetting $z_i^{\mathcal{S}} = \{z_{ij} | x_j \in \mathcal{S}\}$. Obtaining seed forgetting typically takes only a few seconds by running inference on $\mathcal{S}$ given the model fine-tuned on $T_i$ (or fine-tuned for a few steps on $T_i$, which we evaluate separately). We then predict forgetting of the rest $10k - 100k$ upstream examples. Figure 4 illustrates an example of the train-test partition, seed forgetting, and the forgetting to be predicted. We use Root Mean Squared Error (RMSE) or F1 over the $\mathcal{T}_{\text{test}}$ as the metrics of predicting forgetting, *i.e.,* log-perplexity increase or exact match drop.

**Matrix completion approaches**. We run matrix completion algorithms including additive linear models, matrix factorization (MF), and k-nearest neighbors (KNN) models. The additive linear model approximates forgetting as additive effects of learned tasks $\alpha^{(i)}$ and upstream examples $\beta^{(j)}$ ($\alpha^{(i)} + \beta^{(j)}$). The MF models are introduced earlier in Sec. 3. Given the seed forgetting $z_i^{\mathcal{S}}$ of a task $T_i \in \mathcal{T}_{\text{test}}$, KNN finds tasks from $\mathcal{T}_{\text{train}}$ that have similar patterns of forgetting over the seed upstream

Table 3: RMSE (↓) or F1(↑) of predicting example forgetting over a held-out set of upstream examples after fine-tuning LMs on unseen new tasks. We report average performance over 10 random seed sets ($\mathcal{S}$) of upstream examples with known ground truth forgetting beforehand.

| Model | In-Domain | | | | | Out-of-Domain | | | | |
| | OLMo-1B | OLMo-7B | MPT | OLMo-7B Instruct | | OLMo-1B | OLMo-7B | MPT | OLMo-7B Instruct | |
| Metrics | RMSE | RMSE | RMSE | RMSE | F1 | RMSE | RMSE | RMSE | RMSE | F1 |
|---|---|---|---|---|---|---|---|---|---|---|
| *Embedding* | | | | | | | | | | |
| SBERT | 2.81 | 7.44 | 13.86 | 13.94 | 55.46 | 3.53 | 6.16 | 10.59 | 41.22 | 42.95 |
| Text-Embedding-3 | 3.26 | 7.86 | 14.35 | 12.92 | 57.16 | 3.39 | 6.31 | 11.01 | 41.12 | 43.14 |
| *Forgetting Statistics* | | | | | | | | | | |
| Additive | 2.81 | 7.40 | 13.33 | 15.57 | 55.81 | **2.81** | 5.83 | 10.02 | 38.90 | 43.57 |
| KNN | **2.79** | 7.33 | 12.80 | 14.30 | 56.87 | 2.84 | 5.83 | 7.71 | **38.77** | **44.11** |
| MF | 2.80 | **7.14** | **10.41** | **13.74** | **58.16** | 2.82 | **5.76** | **7.03** | 40.47 | 42.91 |

examples $\mathcal{S}$. KNN computes an average of forgetting of top-k similar tasks from $\mathcal{T}_{\text{train}}$ weighted by their similarity as the prediction of forgetting caused by $T_i \in \mathcal{T}_{\text{test}}$ on the upstream examples $x_{1..N}$.

**Comparators of predicting forgetting.** We adapt a prior approach by Jin & Ren (2024) that leverages learned similarity between learned tasks and upstream examples by a LM to predict forgetting. We encode upstream examples $x_j$ and the learned examples $x_i^{1..M_i} \in T_i$ with a frozen pretrained sentence embedding model $h(\cdot)$ followed by two trainable MLP layers to obtain their representations. The final prediction is made with the inner products of two representations $\langle h(x_j), \frac{1}{M_i} \sum_{M_i} h(x_i) \rangle$.

We leave the implementation details of matrix completion approaches and the sentence embedding approach in Appendix D.

## 4.2 Mitigating Forgetting with Predicted Forgetting

**Leveraging predicted forgetting for mitigating forgetting**. We examine the practical utility of predicting forgetting as we sparsely replay upstream examples during forgetting following Jin & Ren (2024); de Masson D'Autume et al. (2019); Ibrahim et al. (2024). We replay one mini-batch of upstream examples every 8 or 32 training steps while fine-tuning on a new task. We perform targeted mitigation of forgetting by prioritizing examples that are predicted to suffer more from forgetting. This is achieved with weighted sampling of upstream examples $x_j$ proportional to $\exp\left(\hat{z}_{ij}/\tau\right)$, where $\hat{z}_{ij}$ is the predicted forgetting and $\tau$ is a temperature hyperparameter.

As we have discussed in Sec. 4.1, predicting forgetting with matrix completion requires seed forgetting $z^S$ to be evaluated. We consider an offline and an online variant of the approach. The *offline* variant performs a replay-free run of fine-tuning on the task $T_i$, after which the seed forgetting will be evaluated. We then perform another run of fine-tuning while replaying examples with the predicted forgetting. This creates computational overhead equivalent to one extra run of fine-tuning, but is still efficient when the training set of fine-tuning is considerably smaller than the upstream data. The *online* variant instead replays random examples for the first 10% of the fine-tuning steps, after which it evaluates seed forgetting and determine examples to be replayed in the rest of the 90% steps. Compared to the offline variant, this mitigates the extra overhead of fine-tuning by trading off the prediction accuracy of forgetting. We illustrate the two variants in Figure 4.

**Baselines of mitigating forgetting.** We compare with various strategies to select upstream examples for sparse replay. We primarily examine whether weighted sampling with predicted forgetting statistically significantly improves over random sampling of upstream examples (Rand). We also compare with a variant of Maximally Interfered Retrieval (MIR-T) (Aljundi et al., 2019a), a selection strategy sharing the similar notion of importance that forgotten examples should be selected for replay. The approach performs bi-level sampling by selecting the most forgotten examples from a small random subset of upstream data. We extend the approach to select forgotten examples after a full training run on a task, instead of single steps, which achieves better performance. In addition, we apply strategies that consider different definitions of upstream example importance. We examine a strategy based on perplexity thresholds (PPL) (Marion et al., 2023), which samples upstream data of which the perplexity is around the median of the distribution. For OLMo-1B, we also sample replayed examples proportional to the gradient inner products (Grad-Prod) (whose correlation with

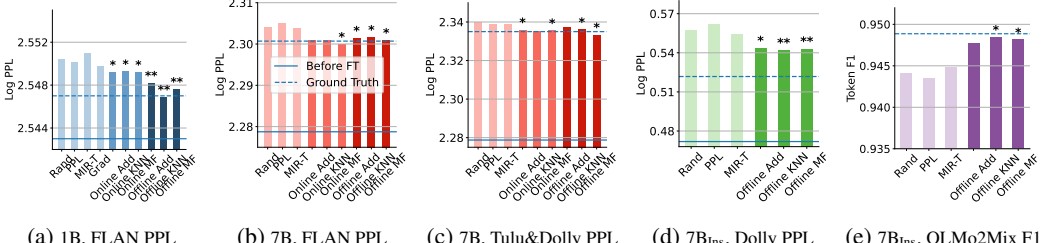

| (a) 1B, FLAN PPL | (b) 7B, FLAN PPL | (c) 7B, Tulu&Dolly PPL | (d) 7B_Ins, Dolly PPL | (e) 7B_Ins, OLMo2Mix F1 |

Figure 5: Log perplexity ($\downarrow$) or Token F1 ($\uparrow$) over upstream data by replay example selection strategies. The solid horizontal lines indicate the log perplexity before fine-tuning (*i.e.,* no forgetting). The dashed lines show the results achieved by upweighting upstream examples according to their actual forgetting after fine-tuning without replay. * and ** indicate significance of improvement ($p < 0.05$ or $p < 0.005$) compared to replaying random examples in paired $t$-tests on all fine-tuning tasks.

forgetting is evaluated in Table 2 in Sec. 3.3), a representative coreset selection approach that utilizes gradient information (Park et al., 2023; Xia et al., 2024). As a reference, we also experiment with upweighting upstream examples with ground truth forgetting $z_{ij}$, which, however, is highly inefficient and repetitive to obtain in practice.

**Metrics.** We measure log-perplexity increase or Token F1 (a softer metric than exact match (Rajpurkar et al., 2016)) drop over a held-out subset of 10,000 examples from the upstream data. This ensures none of the test examples are selected for replay by any of the example selection strategies.

## 4.3 Results of Predicting and Mitigating Forgetting

**Results of predicting example forgetting**. Table 3 summarizes the error of predicting example forgetting over the tasks from the in-domain and out-of-domain test splits. We see matrix completion approaches consistently outperform the sentence embedding model adapted from the prior work. Among the three matrix completion approaches, we notice that MF models in general achieve the lowest prediction error. Besides, KNN in general outperforms additive linear and the embedding model while being highly computationally efficient.

**Mitigating forgetting with the predicted forgetting.** We leverage the online or offline predicted forgetting by the matrix completion algorithms to reweigh examples during replay following the procedure introduced in Sec. 4.2. Figure 5 summarizes log perplexity or Token F1 over the held-out (never replayed) upstream data as we apply different upstream example selection approaches. We notice that example selection based on gradient inner products (Grad) or perplexity threshold (PPL), mainly applied for identifying important training data for a task in prior works, does not show improvement in mitigating forgetting compared to replaying random examples. This implies that example importance defined in these works are different from how easily the examples are forgotten. We also notice that MIR-T does not improve over random sampling in our setup, likely because of the small size of the retrieval candidates relative to the upstream examples. Upweighting examples

Table 4: Computational cost of replay-based approaches as a summation of fine-tuning costs $FT(\cdot)$, inference costs over upstream examples $EV(\cdot)$, and matrix completion costs $MC$. Costs that are minor are displayed in smaller fonts.

| Method | Cost |
|---|---|
| Random | $FT(Y)$ |
| Ground Truth | $2FT(Y) + EV(N)$ |
| Offline MF | $2FT(Y) + EV(S) + MC$ |
| Online MF | $FT(Y) + EV(S) + MC$ |
| MIR-T | $2FT(Y) + Y \cdot EV(S)$ |
| PPL,GradProd | $FT(Y)$ |

with ground truth forgetting (GT) consistently reduces forgetting compared to random examples, shown in dash lines in Figure 5.

By utilizing predicted forgetting by offline additive linear, KNN, and MF models, we statistically significantly reduce forgetting compared to random examples. The MF model achieves statistical significance in most the scenarios, which aligns with its top average prediction performance of example forgetting in Table 3. Utilizing online-predicted forgetting also statistically significantly improves over replaying random examples in 3 of the setups.

**Computational efficiency discussions.** Table 4 summarizes the computation cost of the approaches as a function $FT(\cdot)$ of fine-tuning steps, a function $EV(\cdot)$ of upstream examples whose perplexity is evaluated, and the cost of matrix completion (MC) that is much smaller than LLM inference or training. We note the total number of upstream examples as $N$, the size of seed examples as $S$, and the number of fine-tuning steps as $Y$. As $S$ is much smaller than $M$, majority of computational costs arise from fine-tuning $FT(Y)$ and upstream data evaluation $EV(N)$. Replaying with ground truth forgetting is the most costly, as it introduces inference over potentially very large-scale upstream data. The offline prediction and replay approach saves computations in the scenarios of small fine-tuning datasets and massive upstream data, which is often true in practice. Online prediction of forgetting does not incur extra cost of fine-tuning or upstream data inference, and thereby always being efficient.

**Effect of replay new task learning and patterns of forgetting.** We show in Appendix F that targeted replay also slightly improves new task learning perplexity. Appendix H demonstrates mild change in the pattern of forgetting in the held-out upstream examples when replay is applied.

**Further discussions.** We evaluate fine-tuned OLMo-7B and OLMo-7B-Instruct on unseen LLM leaderboard tasks and present the results in Appendix F and visualize the corresponding forgetting association matrix in Fig. 11. Although we observe slightly improved performance of replaying predicted examples to random or no replay on most forgotten tasks, we do not see statistical significance. Future works can mitigate downstream task forgetting with predicted forgetting with alternative algorithms that leverage past examples (Aljundi et al., 2017; Buzzega et al., 2020). Our results in Appendix I show that matrix completion and embedding based methods can be combined to better improve forgetting prediction performance. In Appendix J, we examine patterns of forgetting by fine-tuning over noisy or adversarial training data.

# 5 Related Works

**Factors that affect forgetting**. In this paper, we primarily studied how the associations between learned and forgotten examples inform forgetting. Prior works have studied various factors that affect forgetting of the models, such as (1) type and size of the LM (Mehta et al., 2021; Scialom et al., 2022; Kalajdzievski, 2024; Mirzadeh et al., 2022; Ramasesh et al., 2022) (2) trainable parts of the model (*e.g.*, LoRA, soft prompts, or full-model tuning) (Biderman et al., 2024a; Razdaibiedina et al., 2023) (3) hyperparameters such as learning rate (Ibrahim et al., 2024; Winata et al., 2023), dropout (Goodfellow et al., 2014), number of training steps (Biderman et al., 2024b; Kleiman et al., 2023) (4) optimizer (Lesort et al., 2023) and training algorithms (*e.g.*, various continual learning algorithms) (Smith et al., 2022; Wang et al., 2022; Shi et al., 2024; Wu et al., 2024), (5) the upstream examples or the knowledge themselves (Toneva et al., 2019; Zhang & Wu, 2024). Future works can study how these factors affect the predictability of forgetting. We consider empirical and theoretical study on the effect of task similarity on forgetting to be most relevant to ours. Ostapenko et al. (2022) empirically study relationships between task similarity and forgetting in foundation models over a sequence of newly learned tasks; our work instead focuses on forgetting of upstream data of LLMs. Theoretical study by Doan et al. (2020); Ding et al. (2024); Evron et al. (2022) dissects effects of the learned tasks on forgetting in linear models or around model initialization. We believe empirical study (Wang et al., 2023; Li et al., 2024a; Zheng et al., 2025) and interpretations of forgetting (Tao et al., 2023; Zhao et al., 2023; Kotha et al., 2024) are complementary to ours and can potentially explain in the future why the associations in $Z$ are often simple, and in which circumstances the associations become more complicated.

**Data selection and data attribution.** Related to our work, data attribution studies faithful algorithms to find training examples that account for a prediction (Koh & Liang, 2017; Ilyas et al., 2022) from a pool of training examples. Park et al. (2023); Xia et al. (2024); Li et al. (2024c); Liu et al. (2024) study the problem of selecting a subset of training data that maximizes performance on a given domain or task at a fixed budget for LLMs. Feldman & Zhang (2020); Tirumala et al. (2022); Biderman et al. (2024b); Swayamdipta et al. (2020) identify memorized, important, or forgetful training data. However, the notion of data importance in these works is different from how likely the upstream examples will be forgotten during fine-tuning. Furthermore, a systematical study on how such importance is dependent on newly learned tasks is still absent. Prior works represented by Aljundi et al. (2019a); Wang et al. (2024); Aljundi et al. (2019b); Huang et al. (2024); Sun et al. (2019) study example selection or synthetic example generation strategies for replay-based continual learning algorithms.

**Predicting model behaviors.** LLMs can display a hybrid pattern of unpredictable to highly predictable behaviors (Ganguli et al., 2022; Wei et al., 2022). Ye et al. (2023); Xia et al. (2020); Schram et al. (2023) study prediction of task performance across datasets and training setups. We perform prediction at the example level which is more fine-grained and under-explored.

## 6 Conclusions

In this paper, we empirically analyzed the associations between learned and forgotten examples in LM fine-tuning. We showed that simple low rank patterns are dominant in the example associations and compared the complexity of the associations across model types and sizes. We showed the example associations alone offer useful information to predict example forgetting when fine-tuning LMs on new tasks. We demonstrated the practical utility of our analysis by showing reduced forgetting as we reweight examples for replay with predicted forgetting. We discuss limitations and future work in Appendix A.

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

# A  Limitations

Although our work aims to be extensive, we consider there are several limitations that can be taken as future work.

**Forgetting under sequential task learning**. In this paper, we analyze and predict forgetting of models that are separately fine-tuned over diverse downstream tasks. We did not explore forgetting under continual or sequential task learning. Although we consider time series forecasting to be relevant to this generalized forgetting prediction problem, we decide to leave it as future work.

**The scope of the definition of forgetting.** We mainly measure forgetting in log perplexity increase or exact match over upstream training data of LLMs. The choice of the metrics is based on the considerations that (1) they are connected to theoretical approximations of forgetting with first-order techniques (Sec. 3.3), allowing proper comparison, and (2) we aim to analyze forgetting on upstream examples that the models have seen before and minimize confounders such as generalization effects on unseen data.

Nevertheless, we note that the forgetting can be defined in many possible ways, each with different focus. Some measures of forgetting, such as an aggregated number of performance drop on domain-specific downstream tasks, are often more sensible to downstream LLM users. Some existing works also separate out "shallow forgetting" of language models, where the forgetting is easily recovered by twisting the prompts (Kotha et al., 2024; Chen et al., 2024). Importantly, forgetting defined in different ways may not always align with each other. We again remind the readers of the definition of forgetting used in the paper, and we consider analysis under different definitions of forgetting to be meaningful future work.

**Theoretical analysis of low-rank example associations.** While we conclude that low-rank example associations are prevalent in LLM forgetting from empirical results, we do not have a theoretical guarantee on why example associations must be low rank, and in which cases the example associations become more complicated. We leave a theoretical understanding of low-rank example associations as future work. Besides, we consider that the line of research on mechanistic interpretability of LLMs (Conmy et al., 2023), which dissects subsets of neurons or computation graphs that are crucial for a prediction, can offer meaningful insights for interpreting the low-rank property.

**Scope of the experiment setups.** Our analysis is restricted to supervised fine-tuning of LMs; other learning setups such as reinforcement learning are not studied. Besides, although we try to cover a broad category of newly learned tasks, the coverage may not be extensive. Due to limited computational resources, it was also not possible to repeat experiments under extensive hyperparameter configurations (*e.g.*, learning rate schedulers), which we leave for future work. Nevertheless, we provide analysis under more learning rate and batch size setups in Appendix H.

**Requirement of open access to upstream data.** Our analysis requires open access to upstream data to evaluate forgetting over them. As a result, replication of our analysis can be limited to LLMs trained on open data or in-house data that can be accessed. In addition, similar to almost every replay-based algorithm, our forgetting mitigation algorithm requires access to upstream data.

**Scale of the upstream examples.** Due to computational resource restrictions, we significantly subsampled upstream pretraining corpora of LLMs in our analysis. Although expanding the set of upstream examples would not theoretically affect the low-rank observations, we decided to leave larger scale analysis as future work. Besides, we consider that matrix completion based forgetting prediction permits sparse association matrices, where upstream example forgetting $z_{ij}$ is sparsely collected, which is a feasible path towards generalizing the forgetting prediction framework to larger scale setups.

**Separating out intended forgetting from unintended forgetting.** Aligning with most prior works on continual learning, our experiments focus on the mitigation of forgetting. We consider the goal aligns with our experiment setup, as upstream examples from Dolma and Tulu are carefully curated and cleaned, and the newly learned tasks intend to accumulate knowledge instead of overwrite previous knowledge. However, we highlight that forgetting is not always detrimental; in fact, the models are often expected to forget (unlearn) sensitive, noisy, and outdated examples (Nguyen et al., 2022; Blanco-Justicia et al., 2025). In addition, active forgetting can serve as a strategy to enhance the plasticity of models when learning new knowledge (Chen et al., 2023). We believe that our analysis of forgetting is helpful for future work that performs targeted forgetting of knowledge.

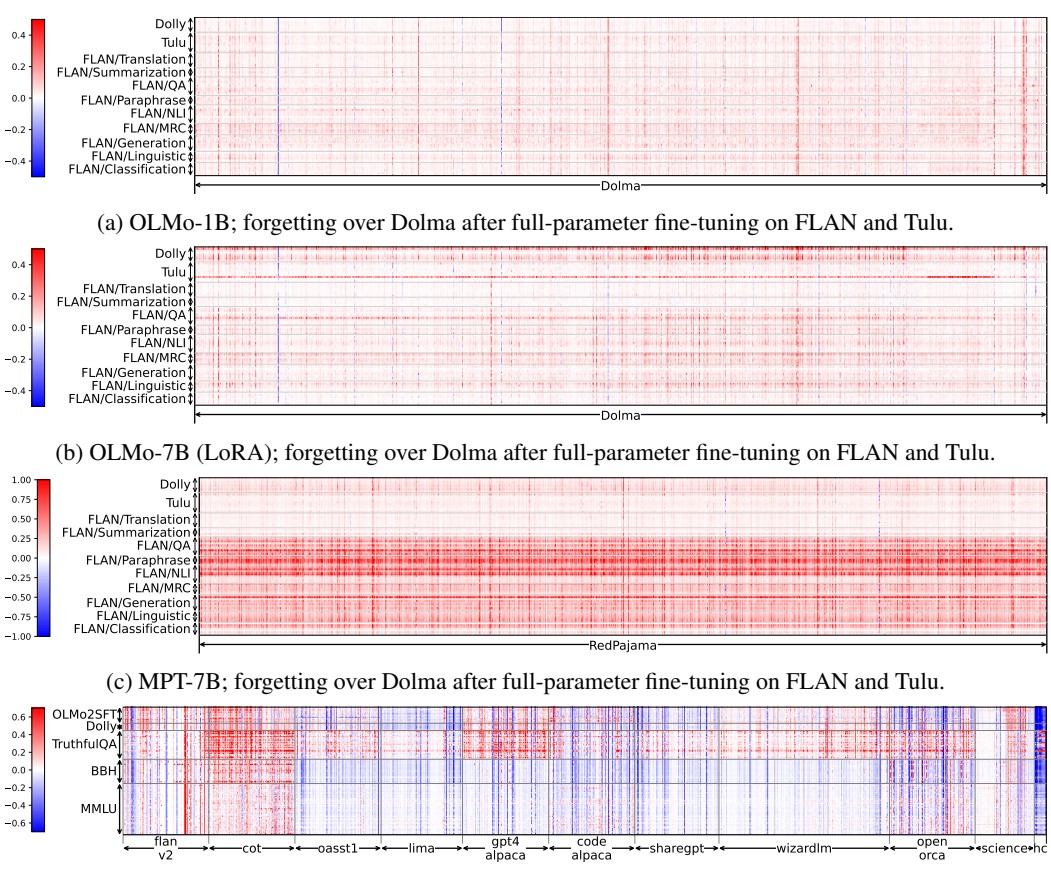

(a) OLMo-1B; forgetting over Dolma after full-parameter fine-tuning on FLAN and Tulu.

(b) OLMo-7B (LoRA); forgetting over Dolma after full-parameter fine-tuning on FLAN and Tulu.

(c) MPT-7B; forgetting over Dolma after full-parameter fine-tuning on FLAN and Tulu.

(d) OLMo-7B-Instruct; forgetting over Tulu after full-parameter fine-tuning on unseen instruction-tuning tasks.

Figure 6: Additional visualized matrices of associations between learned tasks and forgotten examples. We plot forgetting (log-perplexity increase) that occurs on an upstream example (in $x$-axis) after learning a new task (in $y$-axis). Log-perplexity increase can be zero or negative, which indicates no forgetting.

# B Broader Impact

Our research tries to better demystify and mitigate forgetting in LLM fine-tuning. We expect that the better understanding alongside the reduced forgetting can, in turn, encourage model developers to promptly update their LLMs to address limitations of the models and improve their models in continuing efforts. We expect the broader application of the continual learning practice will reduce training costs compared to re-training models, and ultimately result in more powerful models under a controlled training budget.

Although we do not see direct negative impact of predicting example forgetting, we highlight that in real-world continual learning setups, blindly mitigating forgetting may result in outdated knowledge and data privacy breaches in LLMs. Dissecting beneficial and intended forgetting from unintended or catastrophic forgetting requires attention in real-world setups.

# C Dataset, Model, and LM Training Details

**Subsample of upstream datasets**. For OLMo experiments, we sample 141,876 text chunks with length 2,048 from Dolma v1.6-sample as upstream examples. For OLMo-7B-Instruct, we randomly sample an approximately balanced number of examples from each task in Tulu, and filter out examples with input length that exceeds 2,048 (the limit of OLMo models) after tokenization. This results in 10,718 examples. For OLMo2, we sample 70,000 text chunks with length 2,048 from OLMo2-Mix.

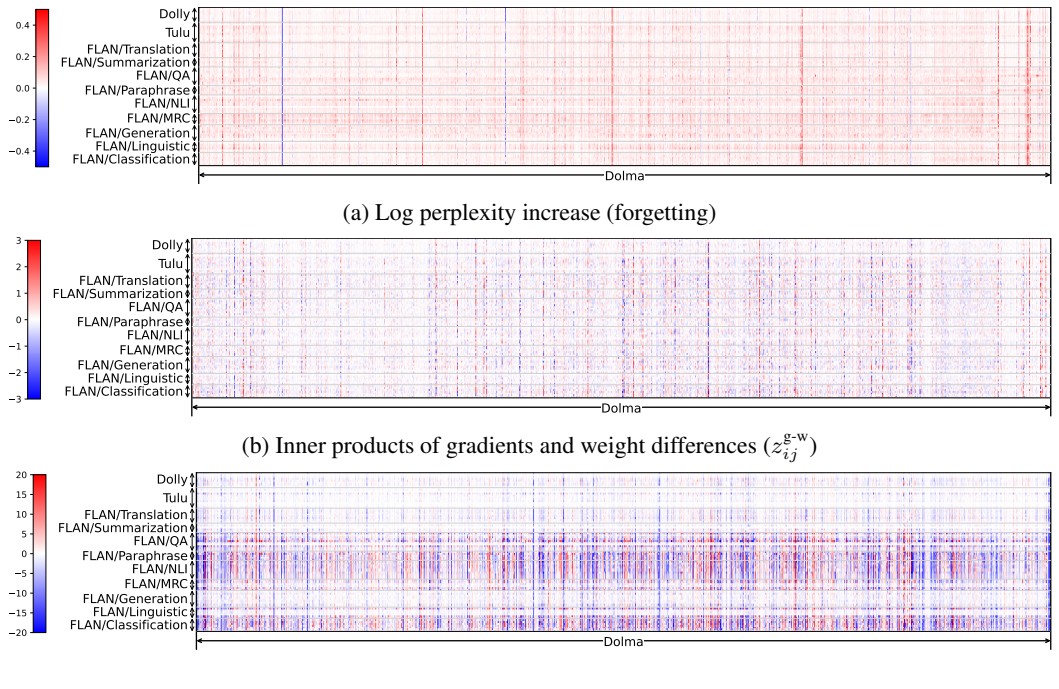

(a) Log perplexity increase (forgetting)

(b) Inner products of gradients and weight differences ($z_{ij}^{\text{g-w}}$)

(c) Negated inner products of gradients (-$z_{ij}^{\text{g-g}}$)

Figure 7: A side-by-side comparison between the matrices of forgetting, inner products of gradients and weight differences ($z_{ij}^{\text{g-w}}$), and the negated inner products of gradients (-$z_{ij}^{\text{g-g}}$) we examined in Sec. 3.3.

For MPT and Pythia, we sample 10,000 2,048-token text chunks from RedPajama and the Pile respectively.

**Learned new tasks and their categorization.** We summarize the list and the categorization of newly learned tasks in Tables 15 and 16. We also annotate tasks used as in-domain training or test tasks.

**Training and evaluation details.** For full-parameter fine-tuning of non-instruction-tuned LLMs of all types, we train the model for 1,000 steps with an effective batch size of 8 and a linearly decaying learning rate of $2e^{-6}$. The learning rate is chosen among $\{1e^{-6}, 2e^{-6}, 5e^{-6}\}$ that achieves the the best average validation perplexity after fine-tuning OLMo-7B on 5 randomly chosen tasks from FLAN. For OLMo-7B-Instruct and MMLU, BBH, TruthfulQA and Dolly, considering the small size of the training sets, we train the models only for 100 steps with an effective batch size of 8. For OLMo2-SFT-Mix tasks, we train the model for 1,000 steps. We use HuggingFace Transformers library for training and VLLM library for efficient inference. Due to computational resource limitations, the statistics of forgetting are collected in a single run.

**Dataset and licenses**. MMLU, BBH, and the Pile are released under MIT license. Truthful QA, Dolma, Redpajama, OLMo models, OLMo2 models, Pythia models, and MPT models are released under Apache 2.0 license. Tulu V2, OLMo2-Mix, and OLMo2-SFT-Mix are released under ODC-By license. Dolly is released under CC BY-SA 3.0 license.

**Computational Infrastructure**. We used 4 Quadro RTX A6000 GPUs for fine-tuning LLMs, and used 1 Quadro RTX A6000 GPU for LLM inference.

# D   Details of Forgetting Prediction and Replay

**Data Splits for Predicting Example Forgetting.** We mark the tasks used as in-domain test splits for predicting example forgetting (Sec. 4) in Tables 15 and 16. The train-test split for the in-domain tasks is randomly generated.

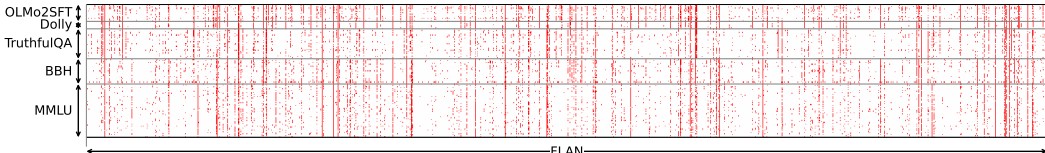

Figure 8: Visualized matrices of associations between learned tasks and forgotten examples on FLAN after fine-tuning OLMo-7B-Instruct, measured with binary exact match drop. Each colored pixel ($z_{ij} = 1$) indicates forgetting of an upstream example $x_j$ after fine-tuning the model on the task $T_i$.

Table 5: Mean and standard deviation of forgetting, and the percentage where the forgetting $z_{ij}$ is non-negative, aggregated over the matrix $Z$. The table uses the same experiment setup as Figure 3.

| Models / Statistics | Avg. Forgetting | Std. Forgetting | Non-Neg % |
|---|---|---|---|
| OLMo-1B | 0.0261 | 0.0425 | 90.36 |
| OLMo-7B | 0.0804 | 0.0626 | 96.29 |
| MPT-7B | 0.0917 | 0.0870 | 98.32 |
| OLMo-7B-Instruct | 0.0637 | 0.3700 | 49.61 |
| OLMo2-7B | 0.0045 | 0.0086 | 81.77 |
| OLMo2-13B | 0.0038 | 0.0078 | 80.11 |
| Pythia-1B | 0.1248 | 0.1406 | 98.53 |
| Pythia-3B | 0.0425 | 0.0523 | 89.24 |
| Pythia-6.9B | 0.1408 | 0.1517 | 99.20 |
| Pythia-12B | 0.1555 | 0.1691 | 99.03 |

**Training and evaluation details.** For MF, we set the dimension of the learnable features (rank) as 5. We train the regression models for 1,000 epochs over the association matrices.

For in-domain test splits, we randomly sample 30 upstream examples and assume the ground truth forgetting is known for these examples. This is required for predicting forgetting on the rest of upstream examples by additive linear, MF, and KNN methods. We repeat the experiment 10 times.

For forgetting prediction based on SBERT embedding similarity, we used all-distilroberta-v1 as the encoder. We also experiment with text-embedding-3-small model by OpenAI. The only trainable part of the prediction models is the MLP layers on top of the text embeddings. We train the models for 2,000 epochs. We note that at inference time the approach does not require ground truth forgetting of a small number of examples.

**Sensitivity analysis to the rank in MF models.**
Table 6 summarizes the RMSE of forgetting prediction with MF models across different ranks $r$. The performance increases with $r$ in the beginning and then drops, indicating an overfit.

Table 6: RMSE of forgetting prediction with matrix factorization (MF) models under different ranks.

**Replaying upstream examples in fine-tuning.**
We sparsely replay 1 mini-batch of 8 upstream examples every 32 steps of model updates while fine-tuning on new tasks. An exception is fine-tuning of OLMo-7B-Instruct models on Dolly, where we perform a replay every 8 steps given the smaller number of model training steps. Given predicted or ground truth forgetting

| Model / Rank | 1 | 3 | 5 | 8 | 10 |
|---|---|---|---|---|---|
| OLMo-7B | 7.31 | 7.17 | 7.14 | **7.12** | 7.14 |
| OLMo-7B-OOD | 6.09 | 5.78 | **5.76** | 5.77 | 5.77 |
| OLMo-1B | 2.85 | 2.82 | 2.80 | **2.78** | **2.78** |
| OLMo-1B-OOD | 2.86 | 2.83 | **2.82** | **2.82** | **2.82** |

$z_{i,1..N}$ on upstream examples $x_{1..N}$ when learning a new task $T_i$, we sample upstream examples to replay from a categorical distribution where $p(x_j) \propto \exp(z_{i,j}/\tau)$, where $\tau$ is a temperature hyper-parameter set as 0.1. The hyperparameter $\tau$ is tuned on a single validation task while reweighting replay examples with the ground truth forgetting $Z$.

# E Details of the Example Similarity Metrics

In this section, we detail the example similarity metrics applied in our analysis in Sec. 3.3.

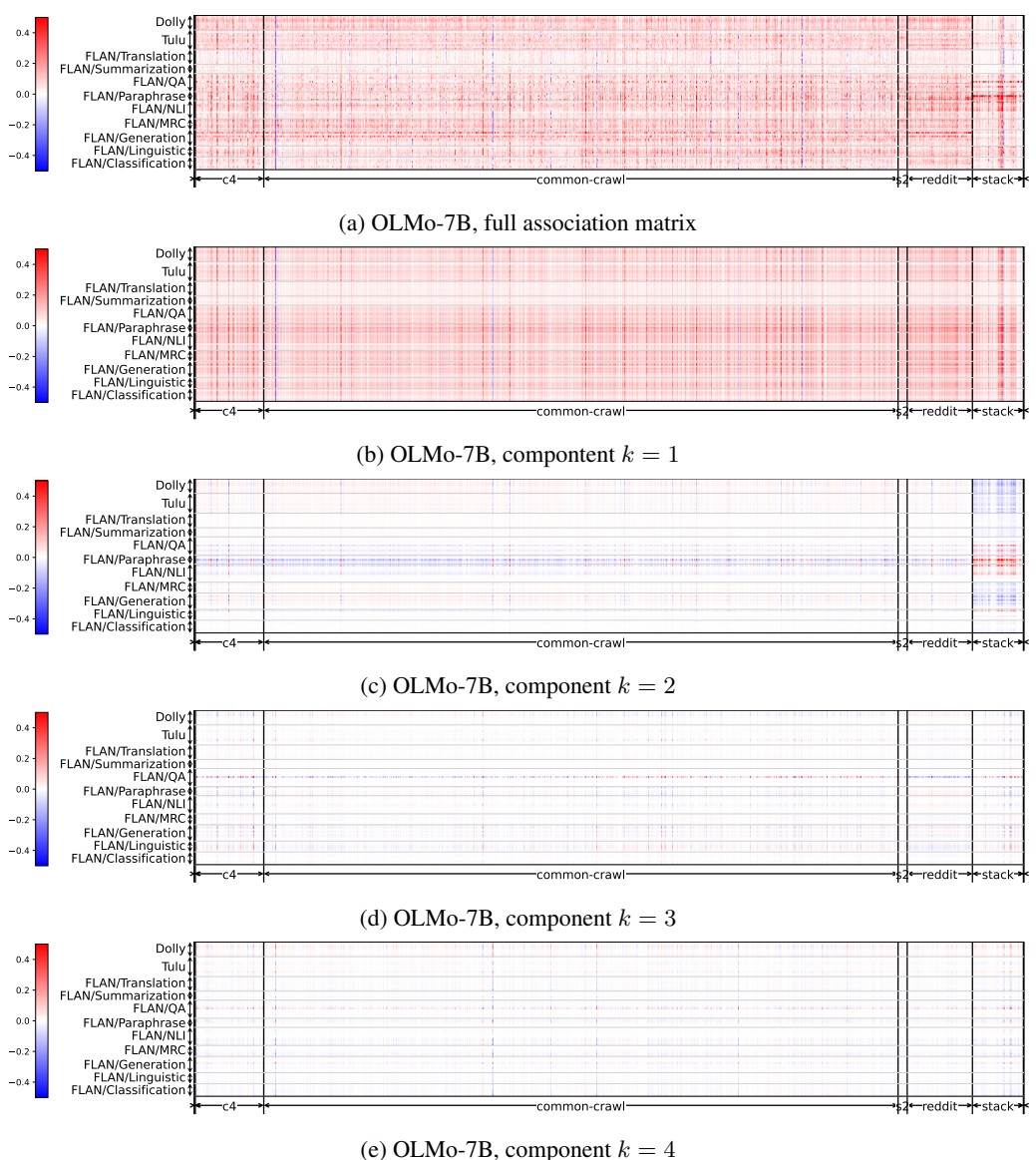

(a) OLMo-7B, full association matrix

(b) OLMo-7B, compontent $k = 1$

(c) OLMo-7B, component $k = 2$

(d) OLMo-7B, component $k = 3$

(e) OLMo-7B, component $k = 4$

Figure 9: Decomposition of $Z$ in OLMo-7B experiments with $k$-th singular value and vectors. Components of higher values of $k$ capture finer-grained details in $Z$.

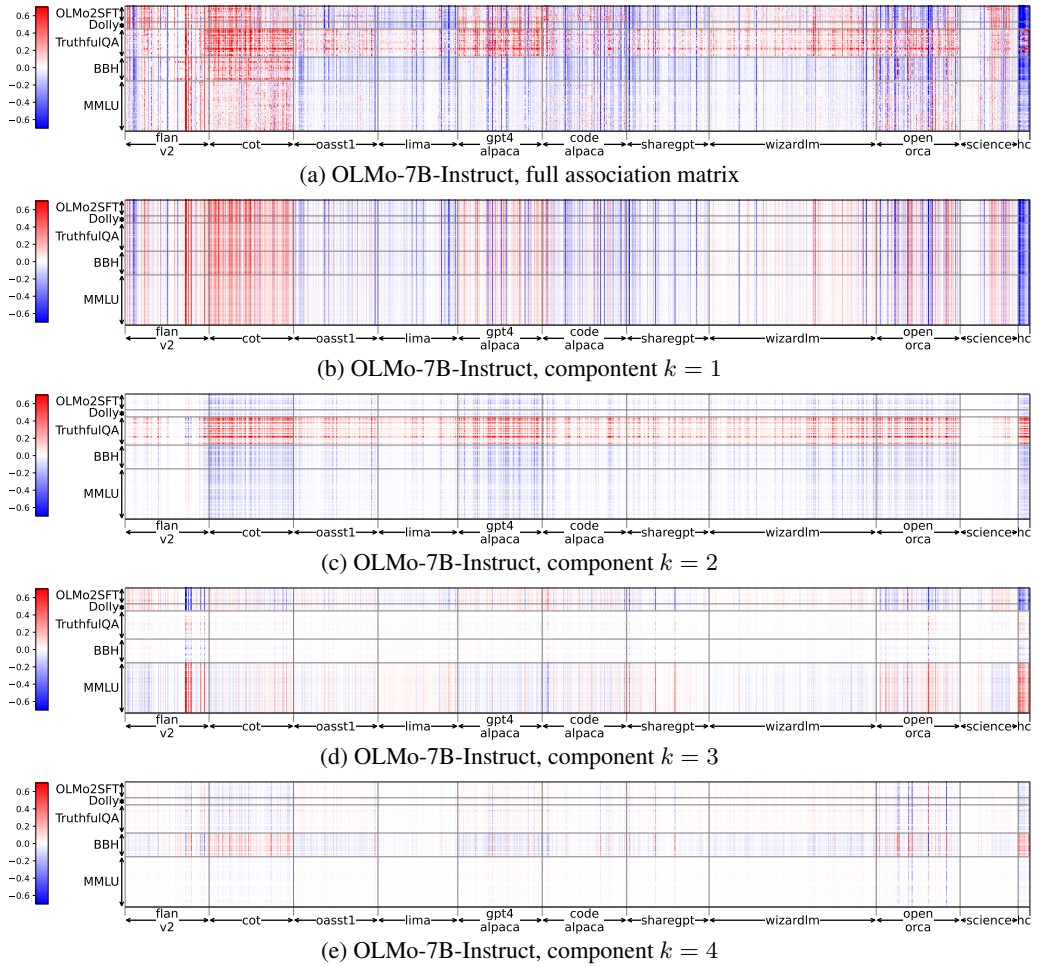

Figure 10: Decomposition of log-perplexity measured forgetting association matrix $Z$ in OLMo-7B-Instruct experiments with $k$-th singular value and vectors. Higher values of $k$ capture finer-grained details in $Z$.

**Token similarity.** We calculate the textual cosine similarity between the learned tasks and upstream examples using TF-IDF vectorized representations of training examples in each learned task $T_i$ and an upstream example $x_j$.

**Representational similarity**. We measure textual representation cosine similarity given by the (1) final layer representations of OLMo-1B, (2) all-distilroberta-v1 sentence transformer (SBERT) model (Reimers & Gurevych, 2019), and (3) OpenAI text-embedding-3-small model. We average representations of maximum 100 training examples as the representation of a task $T_i$.

**Inner products between projected gradients and model weight updates**. The increase of the log perplexity $z_{ij}$ can be approximated with inner products $z_{ij}^{\text{g-w}} = \langle \nabla_\theta f(x_j), \theta_{T_i} - \theta_0 \rangle$ under first-order Taylor expansion (Lee et al., 2019; Doan et al., 2020), where $\nabla_\theta f(x_j)$ is the gradient of the loss of $x_j$ at the initial model before fine-tuning, and $\theta_{T_i} - \theta_0$ are the updates in the model weights after fine-tuning. Following Park et al. (2023); Xia et al. (2024), we use a random projection matrix $P \sim \mathcal{N}_{|\theta| \times d}(0, 1)$ to reduce the dimension of the gradients or the weight changes to save the cost of storing pre-computed statistics, which preserves the inner products with high probability (Johnson & Lindenstrauss, 1984).

**Inner products between projected gradients**. We also measure the negative inner products of the loss gradients between the upstream example $x_j$ and a learned task $T_i$, $-z_{ij}^{\text{g-g}} =$

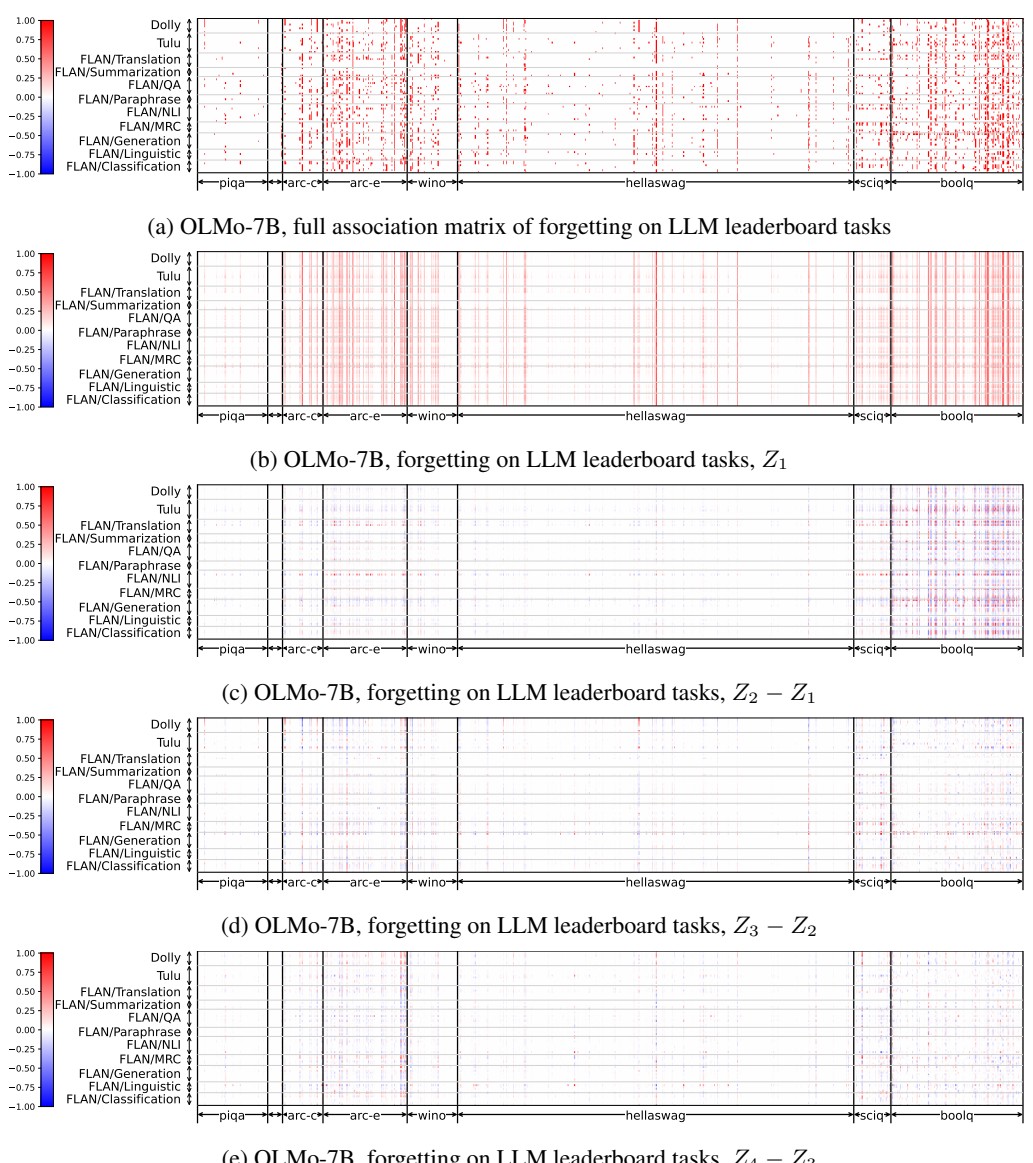

(a) OLMo-7B, full association matrix of forgetting on LLM leaderboard tasks

(b) OLMo-7B, forgetting on LLM leaderboard tasks, $Z_1$

(c) OLMo-7B, forgetting on LLM leaderboard tasks, $Z_2 - Z_1$

(d) OLMo-7B, forgetting on LLM leaderboard tasks, $Z_3 - Z_2$

(e) OLMo-7B, forgetting on LLM leaderboard tasks, $Z_4 - Z_3$

Figure 11: The forgetting association matrix $Z$ between $M = 85$ learned tasks and $N = 1,000$ examples from the LLM leaderboard tasks. The forgetting is defined as a prediction turning from correct to incorrect after fine-tuning. We further show the differences $Z_r - Z_{r-1}$ between rank-$r$ and rank-$r - 1$ approximations. We note that $Z$ is binary, and we define the rank $r$ approximation $Z^r$ as $\sigma(\sum_{k=1}^{r} \boldsymbol{\alpha}_k \boldsymbol{\beta}_k^T)$, where $\sigma$ is the sigmoid function.

$-\langle \nabla_\theta f(x_j), \nabla_\theta f(T_i) \rangle$, as an approximation of forgetting (Lopez-Paz & Ranzato, 2017; Chaudhry et al., 2019).

## F  Effect of Targeted Replay on New-Task Learning and Downstream Performance

**Targeted replay does not impede learning of new tasks, as shown with decreased validation perplexity.** Table 7 summarizes log perplexity on the validation set of learned tasks after fine-tuning the models with different replay strategies. We notice that example replay and targeted replay often decrease validation perplexity of new tasks. This improvement of new task perplexity can be attributed

Table 7: Validation log perplexity of the learned tasks after fine-tuning LMs with different replay strategies. *: $p < 0.05$ improvement compared to no replay.

| Model | OLMo-1B | OLMo-7B | OLMo-7B | OLMo-7B-Instruct |
|---|---|---|---|---|
| Dataset | FLAN | FLAN | Tulu & Dolly | OLMo2-SFT-Mix |
| No Replay | 0.9723 | 0.7736 | 1.6397 | 0.6677 |
| Random | 0.9621 | 0.7691 | 1.6452 | **0.6671** |
| MF-Offline | **0.9601**$^*$ | **0.7684**$^*$ | **1.6294**$^*$ | 0.6674 |

Table 8: Downstream task performance of OLMo-7B models before and after fine-tuning on dolly tasks.

| | ARC-Easy | ARC-Challenge | Boolq | Hellaswag | Openbookqa | Piqa | Sciq | Winogrande |
|---|---|---|---|---|---|---|---|---|
| Metrics | Acc-norm | Acc-norm | Acc | Acc-norm | Acc-norm | Acc-norm | Acc-norm | Acc |
| Before FT | 68.77 | 40.36 | 72.41 | 75.65 | 42.20 | 79.54 | 88.60 | 66.29 |
| No Replay | 67.34 | 42.28 | 74.82 | 76.89 | 44.65 | 80.05 | 84.09 | 67.89 |
| Random | 67.48 | 42.43 | 74.33 | 77.26 | 44.88 | 79.97 | 84.77 | 67.33 |
| MF-Offline | 67.52 | 42.35 | 74.50 | 77.56 | 44.25 | 79.97 | 85.10 | 67.17 |

to less forgetting of general knowledge during fine-tuning. Besides, the results imply that targeted replay does not simply trade off new task learning for reduced forgetting.

**LLM leaderboard task evaluation**. We evaluate the change of performance on Open LLM Leaderboard tasks[1] before and after fine-tuning a model. We use same set of LLM leaderboard tasks as the OLMo technical report (Groeneveld et al., 2024). For OLMo-7B models, we evaluate downstream performance of 8 models fine-tuned the Dolly tasks. For OLMo-7B Instruct, We evaluate downstream performance after fine-tuning on 4 held-out tasks from OLMo2-SFT-Mix (marked in Table 9). We compare LLM leaderboard task performance of no replay, replaying random examples, and replaying with forgetting predicted by offline MF. Table 8 summarizes results on OLMo-7B; Table 9 summarizes results on OLMo-7B-Instruct.

We notice that fine-tuning OLMo-7B on Dolly improves downstream performance on most of the downstream tasks. This aligns well with the purpose of fine-tuning a LM that is not instruction-tuned. Nevertheless, we notice performance degradation on two of the tasks, namely ARC-Easy and Sciq, which indicates forgetting. Although offline MF Replay achieves higher accuracy scores on these two tasks compared to no-replay, we do not find the improvement statistically significant.

We see fine-tuning OLMo-7B-Instruct over new tasks without any replay improves metrics scores on some tasks (MUSR, GPQA) but causes forgetting on other tasks (MMLU, BBH, IFEval). Among tasks where performance improved, we do not see benefits of example replay. Among tasks with decreased performance, we see offline-MF mitigates forgetting compared to no replay or random replay only very marginally.

We conjecture that replay-based approaches are not sufficient to significantly mitigate forgetting on their own, and can be combined with other approaches. We leave more effective algorithms to mitigate downstream task forgetting with predicted forgetting as future work.

## G  Towards Interpreting Fine-Grained Associations

We visualize progressive reconstruction with $k$-th singular value and singular vectors for OLMo experiments in Figure 9.

**Semantic meanings of the $k$-th component in the low-rank approximation of the association matrix $Z$.** We perform further analysis into the patterns captured by the $k$-th singular value and singular vectors by identifying the most relevant learned tasks and upstream example domain to the component. For each $k$ and its corresponding component $\alpha_k \beta_k^T$, we extract top 3 rows with the

---

[1]https://huggingface.co/docs/leaderboards/open_llm_leaderboard/about

Table 9: Average downstream task performance of OLMo-7B-Instruct models before and after fine-tuning on 4 held-out OLMo2-SFT-Mix tasks.

| | MMLU-Pro | BBH | IF-Eval | MUSR | GPQA |
|---|---|---|---|---|---|
| Metrics | 5-shot Acc | 3-shot Acc-norm | 0-shot Inst | 0-shot Acc-norm | 0-shot Acc-norm |
| Before FT | 18.20 | 37.65 | 39.93 | 38.31 | 26.35 |
| No Replay | 17.21 | 36.76 | 21.04 | **40.20** | **27.99** |
| Random | 17.37 | 36.85 | 20.97 | 39.77 | 27.73 |
| MF-Offline | **17.43** | **36.89** | **21.14** | 39.50 | 27.69 |

Table 10: Semantic meaning of the $k$-th component in the factorization of the example association matrix $Z$. We identify top relevant learned tasks and upstream example domains to the $k$-th component in the factorization of the example association matrix $Z$.

| | OLMo-7B | | OLMo-1B | |
|---|---|---|---|---|
| | Learned Tasks | Forgotten Domain | Learned Tasks | Forgotten Domain |
| $k = 1$ | flan/paws_wiki
flan/glue_mrpc
flan/story_cloze | None | flan/squad_v2
flan/fix_punct
tulu/open_orca | None |
| $k = 2$ | flan/opinion_abstracts_idebate
dolly/general_qa
flan/story_cloze | StackOverflow (Less) | flan/mnli_matched
flan/mnli_mismatched
flan/snli | None |
| $k = 3$ | flan/story_cloze
flan/fix_punct
flan/true_case | None | flan/squad_v2
flan/quac
flan/fix_punct | None |
| $k = 4$ | math_dataset
dolly/general_qa
flan/opinion_abstracts_idebate | None | flan/rte
flan/opinion_abstracts_idebate
flan/story_cloze | None |

highest mean (*i.e.*, top 3 relevant learned tasks $T_i$). We also extract top 50 columns with highest mean (*i.e.* top 50 relevant upstream examples) and the domain where these upstream examples are drawn from. For OLMo models, the domains are one of C4, common-crawl, Gutenberg books, Reddit, Science, StackOverFlow, and Wikipedia. We compare the distribution of domains in the top 50 upstream examples, and perform a $z$-test to determine upstream example domain that is significantly more or less forgotten compared to a prior domain distribution of top 50 most forgotten upstream examples (columns with highest mean in $Z$). The results are summarized in Table 10.

We highlight some notable patterns in Table 10. (1) Some component $Z_k$ highlights forgetting patterns of upstream examples from certain domains. On OLMo-7B, the second component ($k = 2$, also visualized in Figure 9(c)) highlights patterns where StackOverFlow examples are disproportionally more or less forgotten. (2) Some components $Z_k$ highlight forgetting when learning specific types of tasks. For example, the second component ($k = 2$) on OLMo-1B highlights forgetting patterns after learning NLI tasks (mnli_matched, mnli_mismatched, snli). This also exemplifies how learning similar tasks causes similar sets of upstream examples to be more forgotten.

# H   Effect of Model Training Setups on Forgetting

In this section, we examine patterns of forgetting and its low-rank approximations across various training setups.

**Low-rank approximations hold at various learning rates, batch sizes, and LoRA fine-tuning.**
Table 11 summarizes $R^2$ of low-rank approximations. Reducing learning rate from the default 2e-6 to 1e-6 causes very minor change of $R^2$; Nevertheless, we notice increased $R^2$ under a larger learning rate like 5e-6, indicating even simpler associations between the learned tasks and forgotten upstream examples. Under this overly large learning rate, the model suffers substantially more forgetting on all examples. Besides, increasing the batch size (under the same number of parameter update steps) causes an increase in $R^2$, indicating simpler associations, possibly due to reduced variance of learned models under a larger batch size. LoRA fine-tuning (Hu et al., 2022) results in simpler associations despite the model forgets less than full parameter fine-tuning.

Table 11: $R^2$ of approximating forgetting association matrix $Z$ across different learning rates (LR) and batch sizes (BS). The default learning rate and batch sizes are 2e-6 and 8. We report results at $M = 19$ learned tasks from Tulu and Dolly and $N = 141,871$ upstream examples from Dolma.

| Model | Setting | $r = 2$ | $r = 3$ | $r = 5$ |
|---|---|---|---|---|
| OLMo-7B | Default | 0.6981 | 0.7542 | 0.8232 |
| | LR=1e-6 | 0.7071 | 0.7697 | 0.8361 |
| | BS=32 | 0.8041 | 0.8453 | 0.9011 |
| | LR=5e-6 | 0.9331 | 0.9582 | 0.9718 |
| | LoRA, LR=1e-4 | 0.8963 | 0.9307 | 0.9562 |
| OLMo-1B | Default | 0.8344 | 0.8836 | 0.9177 |
| | LR=1e-6 | 0.8471 | 0.8989 | 0.9214 |
| | BS=32 | 0.8724 | 0.9289 | 0.9607 |

Table 12: Sample Pearson correlation coefficient between upstream example forgetting under different replay strategies. The results are collected with OLMo-7B models fine-tuned on 8 Dolly tasks.

| Setup A | Setup B | Sample Pearson $r$ |
|---|---|---|
| No Replay | No Replay + seed change | 0.9403 |
| No Replay | Random Replay | 0.7919 |
| No Replay | MF-Offline Replay | 0.8121 |
| Random Replay | MF-Offline Replay | 0.7709 |

**Replay can change the patterns of forgetting compared to no replay, but the change is mild.** We compute sample Pearson correlation coefficient between upstream example forgetting collected under different replay strategies, and more specifically (1) no replay (2) random replay and (3) MF-Offline replay on OLMo-7B and 8 Dolly tasks. We evaluate forgetting of upstream examples held-out from replay. We include correlations between upstream example forgetting under two different random seeds during training as the baseline.

We summarize the results in Table 12. The correlations between no replay and replay are around 0.8, lower than the 0.94 baseline, but are still strongly positive. Replay example selection strategies (random or MF-offline) do not have a strong impact on the correlation. The results imply replay changes patterns of forgetting to a mild degree. Future work can study the limit where the patterns of forgetting experience notable changes.

# I  Combining Embedding and Matrix Completion based Prediction

Although our results in Sec. 4 demonstrate subpar performance of the embedding-based approach, we show combining embedding and matrix completion approaches leads to improvement over the both. Specifically, we fit the residual error of the matrix completion model $Z - Z_{MC}$ with the embedding model, where $Z_{MC}$ is the prediction given by the matrix completion algorithms. We use the better

Table 13: Combining matrix completion and embedding based prediction of forgetting. We report RMSE ($\downarrow$) or F1($\uparrow$) of predicting example forgetting over a held-out set of upstream examples after fine-tuning LMs on unseen new tasks.

| Model | In-Domain | | | | | Out-of-Domain | | | | |
|---|---|---|---|---|---|---|---|---|---|---|
| | OLMo-1B | OLMo-7B | MPT | OLMo-7B Instruct | | OLMo-1B | OLMo-7B | MPT | OLMo-7B Instruct | |
| Metrics | RMSE | RMSE | RMSE | RMSE | F1 | RMSE | RMSE | RMSE | RMSE | F1 |
| Matrix Completion | 2.79 | 7.14 | 10.41 | 13.74 | 58.16 | 2.82 | 5.76 | 7.03 | 38.90 | 43.57 |
| SBERT | 2.81 | 7.44 | 13.86 | 13.94 | 55.46 | 3.53 | 6.16 | 10.59 | 41.22 | 42.95 |
| SBERT + MC | **2.76** | **7.09** | **10.20** | **13.25** | **59.36** | **2.76** | **5.75** | **7.00** | 40.36 | 42.98 |

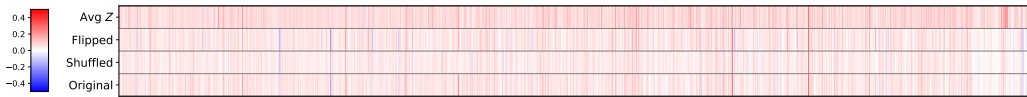

(a) OLMo-7B, forgetting over Dolma after fine-tuning over Ultrafeedback with shuffled or flipped preferred responses.

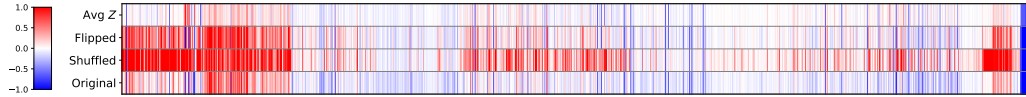

(b) OLMo-7B-Instruct, forgetting over Tulu after fine-tuning over Ultrafeedback with shuffled or flipped preferred responses.

Figure 12: Forgetting after fine-tuning OLMo-7B or OLMo-7B-Instruct on Ultrafeedback dataset with original, shuffled, or flipped preferred responses.

Table 14: Correlations between column mean of $Z$ (in main experiments in Sec. 3, visualized in Figures 2 and 6) and the forgetting occurred on upstream data after fine-tuning on clean, noisy (shuffled) or adversarial (flipped) variants of UltraFeedback.

|  | OLMo-7B | | | OLMo-7B-Instruct | | |
|---|---|---|---|---|---|---|
|  | Pearson $\rho$ | Spearman $\rho$ | Avg. Forgetting | Pearson $\rho$ | Spearman $\rho$ | Avg. Forgetting |
| Original | 0.5884 | 0.4813 | 0.0466 | 0.7382 | 0.7632 | -0.0432 |
| Shuffled | 0.5865 | 0.4591 | 0.0375 | 0.4294 | 0.5624 | 0.5149 |
| Flipped | 0.6535 | 0.5289 | 0.0463 | 0.6516 | 0.6893 | 0.1103 |

of the additive and MF models reported in Table 3. Table 13 summarizes the results. We see that combining the two approaches improves performance over the two.

We consider the embedding based approaches have potential to better capture fine-grained knowledge conflicts that leads to forgetting of one example by learning the other (*e.g.*, two documents stating contradictory facts) that is missed by matrix completion approaches that rely on statistics of forgetting.

# J  Fine-tuning over Noisy or Adversarial Training Data

In this section, we examine forgetting after fine-tuning on clean, noisy, or adversarial new tasks. We experiment with Ultrafeedback (Cui et al., 2024), an instruction tuning dataset where each instruction is paired with one most preferred response and many rejected responses. For the *clean* variant, we fine-tune the model over the preferred responses. For the *noisy* variant, we shuffle the set of the preferred responses so that the instruction and the response mismatch. For the *adversarial* variant, we fine-tune the model over the rejected responses. We leave preference pair fine-tuning that simultaneously utilize both preferred and rejected responses as future work.

Figure 12 visualizes the forgetting on Dolma or Tulu after fine-tuning OLMo-7B and OLMo-7B instruct, represented in a matrix $Z_U \in \mathbb{R}^{3 \times N}$. We also include column mean of $Z$ (*i.e.,* upstream example forgetting averaged over all tasks $T_{1..M}$) in the main experiments in Sec. 3 as references. In Table 14, we present the correlations between $Z_U$ and the column mean of $Z$.

On OLMo-7B, our results suggests that upstream example forgetting after fine-tuning on rejected responses do not correlates less with the task-averaged upstream example forgetting (column mean of $Z$) compared to fine-tuning on the clean data. On OLMo-7B-Instruct, however, we see a clear drop in the correlation by fine-tuning on the flipped data, very likely because the data contradicts the upstream instruction tuning data. Nevertheless, we notice the correlation with the task-averaged forgetting is still strongly positive. Meanwhile, fine-tuning on shuffled data causes forgetting that is least correlated with the task-averaged upstream example forgetting, indicating more significant change in the patterns of forgetting. Still, we notice a strongly positive correlation.

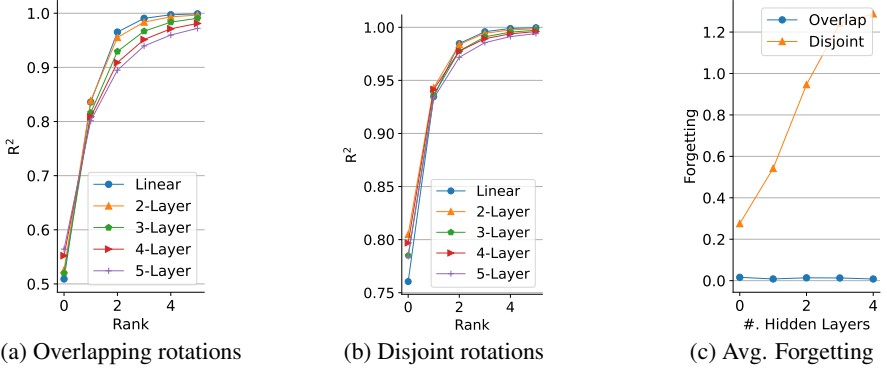

| (a) Overlapping rotations | (b) Disjoint rotations | (c) Avg. Forgetting |

Figure 13: $R^2$ of low-rank approximations of the example association in the forgetting of MLP models on rotated MNIST experiments. We report $R^2$ when the rotation of the newly learned task overlaps or is disjoint with the upstream data separately. $y$-axes are not in the same scale.

## K Synthetic Dataset Experiments

In this section, we present a set of experiments on a synthetic dataset, Rotated-MNIST, broadly used in continual learning research. We aim to provide intuition for future research about how the complexity of the example associations can depend on (1) the coverage of knowledge represented in upstream examples, and (2) the size of the models, in highly controlled setups. We apply a training setup that resembles pretraining and fine-tuning paradigm in transformer language models. Specifically, the models are first pre-trained on 10 rotated variants (0-90°) of the MNIST digit classification dataset (as upstream tasks and examples). Then, the models are fine-tuned on one of 40 unseen rotations for one epoch. Among the 40 unseen rotations, 20 are drawn from the same range as upstream examples (0-90°), while the other 20 are drawn from a disjoint range (-90-0°). This separation controls the amount of shared knowledge between the newly learned and upstream tasks.

We mostly apply other training setups and hyperparameters in Aljundi et al. (2019a). Each task (rotation) includes 1,000 training examples. We train a MLP classifier to predict among 10 digits given an input image without providing its rotation (or the task identifier). We experiment MLP classification models with 1 layer (a linear model) to 5 layers. We collect the example associations $Z$, and visualize them in Figure 14. The upstream and the newly learned rotation tasks are ordered by their rotations in the $x$ or $y$ axis.

**Effects of knowledge coverage.** When the rotations of the newly learned tasks overlaps with the range of upstream examples, the forgetting is harder to approximate with low-rank approximations, resulting in a $R^2$ only around 0.5 for all MLP models with rank-1 approximation. In contrast, the $R^2$ scores are much higher when the rotations do not overlap, alongside higher average forgetting; $R^2$ with rank-1 approximation is higher than 0.8 in these setups. The results imply that the amount of shared knowledge between fine-tuning tasks and upstream examples can have an impact on the complexity of the example associations. In other words, models trained with a broad coverage of upstream examples, or that cover knowledge required for diverse fine-tuning tasks, can yield more complicated example associations. In the context of LLMs, we have noticed that more powerful LLMs (such as OLMo and OLMo2, compared to Pythia and MPT) with broader coverage of knowledge yield more complicated patterns of forgetting.

**Effects of model sizes**. We compare the MLPs of different number of layers trained on the same upstream data. From Figure 13, the patterns of forgetting are nosier in deeper models. The $R^2$ scores at rank 3 or 5 decreases with added layers in MLPs, providing a quantitative measure of increased complexity between learned tasks and forgotten examples.

To summarize, our analysis with synthetic datasets provide intuition about the effect of knowledge coverage and model sizes on the complexity of example associations in forgetting. We hope the set of synthetic experiments can inspire more comprehensive study on how the complexity of example associations are affected by various factors in future work.

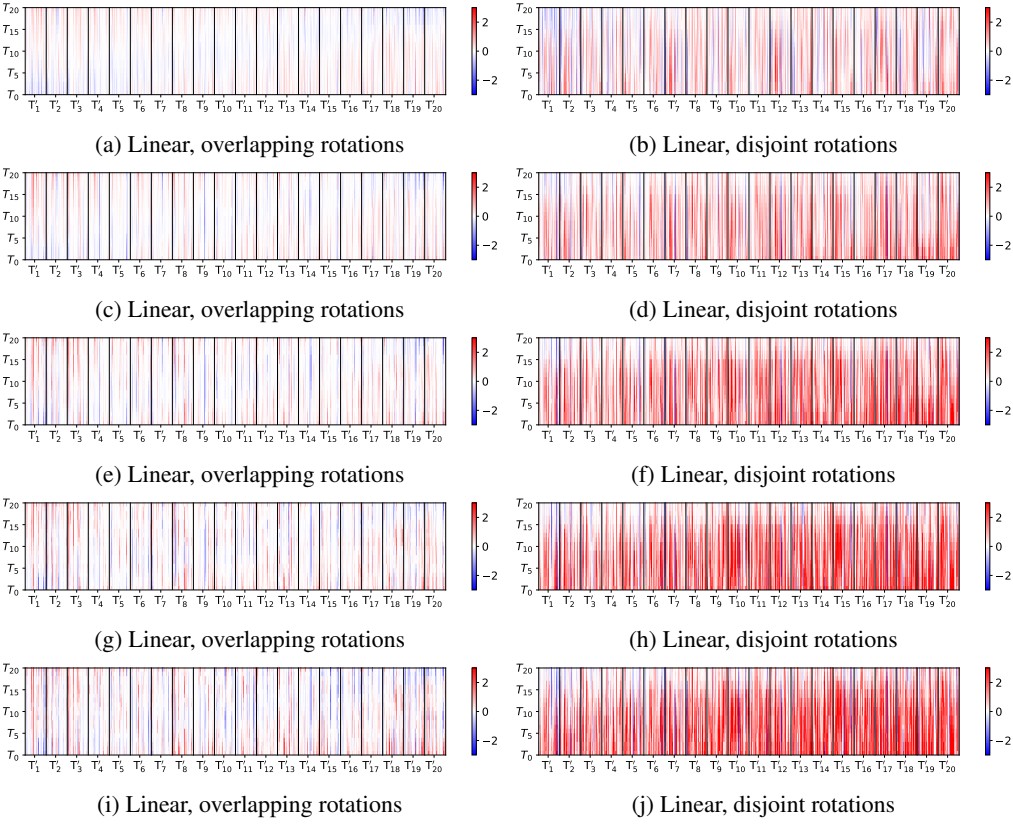

Figure 14: Example associations between learned tasks and forgotten upstream examples on Rotated MNIST with overlapping or disjoint ranges of rotations. We measure increase in the cross-entropy loss as forgetting.

| Task Category | Task | Task Category | Task |
|---|---|---|---|
| FLAN/Classification | aeslc | FLAN/QA | arc_challenge* |
| | ag_news_subset | | arc_easy* |
| | imdb_reviews | | bool_q |
| | sentiment140 | | coqa* |
| | sst2 | | cosmos_qa |
| | trec* | | math_dataset* |
| | yelp_polarity_reviews* | | natural_questions* |
| FLAN/Linguistic | cola | | openbookqa* |
| | definite_pronoun_resolution* | | piqa |
| | fix_punct* | | trivia_qa* |
| | true_case | FLAN/Summarization | cnn_dailymail |
| | word_segment | | gigaword |
| | wsc* | | multi_news |
| FLAN/Generation | common_gen | | samsum |
| | copa | | wiki_lingua_english_en |
| | dart | FLAN/Translation | para_crawl_enes |
| | e2e_nlg* | | wmt14_enfr |
| | hellaswag | | wmt16_translate_csen |
| | opinion_abstracts_idebate* | | wmt16_translate_deen |
| | opinion_abstracts_rotten_tomatoes | | wmt16_translate_fien |
| | story_cloze | | wmt16_translate_roen |
| | web_nlg_en | | wmt16_translate_ruen* |
| FLAN/MRC | drop | | wmt16_translate_tren* |
| | multirc | Tulu | open_orca |
| | quac | | oasst1 |
| | record | | lima |
| | squad_v1 | | code_alpaca |
| | squad_v2 | | gpt4_alpaca |
| FLAN/NLI | anli_r1 | | cot |
| | anli_r2 | | science |
| | anli_r3 | | flan_v2 |
| | cb* | | sharegpt |
| | mnli_matched | | hard_coded |
| | mnli_mismatched | | wizardlm |
| | qnli* | Dolly | brainstorming |
| | rte | | closed_qa |
| | snli | | information_extraction |
| | wnli | | classification |
| FLAN/Paraphrase | glue_mrpc | | open_qa |
| | glue_qqp* | | general_qa |
| | paws_wiki | | creative_writing |
| | stsb | | summarization |
| | wic* | | |

Table 15: The list of learned tasks in our experiments on OLMo-1B, OLMo-7B and MPT-7B. * notes for tasks used as the in-domain test split in forgetting prediction experiments in Sec. 4.

| Task Category | Task | Task Category | Task |
|---|---|---|---|
| MMLU | abstract_algebra | | object_counting* |
| | anatomy | | penguins_in_a_table |
| | astronomy | | reasoning_about_colored_objects |
| | business_ethics | | ruin_names |
| | clinical_knowledge | | salient_translation_error_detection |
| | college_biology* | | snarks |
| | college_chemistry | | sports_understanding |
| | college_computer_science | | temporal_sequences |
| | college_mathematics | | tracking_shuffled_objects_five_objects |
| | college_medicine* | | tracking_shuffled_objects_seven_objects |
| | college_physics | | tracking_shuffled_objects_three_objects |
| | computer_security | | web_of_lies |
| | conceptual_physics* | | word_sorting |
| | econometrics | TruthfulQA | Nutrition |
| | electrical_engineering | | Stereotypes |
| | elementary_mathematics | | Confusion |
| | formal_logic | | Psychology |
| | global_facts* | | Language |
| | high_school_biology* | | Sociology |
| | high_school_chemistry | | Finance |
| | high_school_computer_science | | Indexical Error |
| | high_school_european_history* | | Science |
| | high_school_geography | | Misconceptions |
| | high_school_government_and_politics | | Economics |
| | high_school_macroeconomics | | Education |
| | high_school_mathematics | | Proverbs |
| | high_school_microeconomics | | Conspiracies |
| | high_school_physics* | | Religion |
| | high_school_psychology | | Statistics |
| | high_school_statistics | | Misquotations |
| | high_school_us_history* | | Subjective |
| | high_school_world_history | | Law |
| | human_aging* | | History |
| | human_sexuality* | | Fiction |
| | international_law | | Mandela Effect |
| | jurisprudence | | Politics |
| | logical_fallacies* | | Misinformation |
| | machine_learning | | Logical Falsehood |
| | management* | | Distraction |
| | marketing* | | Weather |
| | medical_genetics | | Myths and Fairytales |
| | miscellaneous | | Superstitions |
| | moral_disputes | | Advertising |
| | moral_scenarios* | | Paranormal |
| | nutrition | | Health |
| | philosophy* | Dolly | brainstorming |
| | prehistory | | rte closed_qa |
| | professional_accounting | | snli information_extraction |
| | professional_law | | wnli classification |
| | professional_medicine* | | FLAN/Paraphrase glue_mrpc open_qa |
| | professional_psychology | | glue_qqp* general_qa |
| | public_relations* | | paws_wiki creative_writing |
| | security_studies | | stsb summarization |
| | sociology* | OLMo2SFT-Mix | coconot* |
| | us_foreign_policy* | | evol_codealpaca_heval |
| | virology | | flan_v2 |
| | world_religions | | no_robots |
| BBH | boolean_expressions* | | numinamath_tir_math* |
| | causal_judgement | | oasst1 |
| | date_understanding | | personahub_code |
| | disambiguation_qa | | personahub_ifdata |
| | dyck_languages* | | personahub_math* |
| | formal_fallacies* | | aya |
| | geometric_shapes | | open_math_2_gsm8k |
| | hyperbaton* | | personahub_math_interm_algebra |
| | logical_deduction_five_objects* | | sciriff |
| | logical_deduction_seven_objects | | synthetic_finalresp_wildguardmixtrain |
| | logical_deduction_three_objects | | table_gpt |
| | movie_recommendation* | | wildchat |
| | multistep_arithmetic_two | | wildjailbreak* |
| | navigate | | personas-math-grade |

Table 16: The list of learned tasks in our experiments on OLMo-7B-Instruct. * notes for tasks used as the in-domain test split in forgetting prediction experiments in Sec. 4.

