# OpenReview forum: "Demystifying Language Model Forgetting with Low-rank Example Associations"
_NeurIPS.cc/2025/Conference — NeurIPS 2025 poster_

### Official Review · Reviewer_s6Xq · 2025-07-02

**Clarity:** 3
**Significance:** 3
**Originality:** 3
**Rating:** 4
**Confidence:** 4

**Summary:**

This paper demystifies LLM forgetting by showing that the associations between newly learned tasks and forgotten upstream examples form a low-rank matrix. Based on this finding, the authors introduce a highly efficient method to predict future forgetting by treating it as a matrix completion problem. The practical utility is confirmed by using these predictions to guide a weighted replay strategy, which reduces forgetting during fine-tuning.

**Questions:**

please see the weakness part.

**Ethical Concerns:**

["NO or VERY MINOR ethics concerns only"]

**Quality:**

3

**Strengths And Weaknesses:**

Strong points：
1. Models catastrophic forgetting with an innovative and analyzable low-rank matrix framework.
2. Formulates forgetting prediction as a matrix completion problem, which is more effective and computationally cheaper than prior methods.
3. Provides extensive empirical evidence across multiple models and scales, with a closed-loop validation showing that predicted forgetting can be used to mitigate actual forgetting.


Weak:
1. In Figure 3, the paper shows that the complexity of forgetting associations differs significantly between OLMo-1B and OLMo-7B, with the latter exhibiting more complex patterns (lower R² at the same rank). Furthermore, Figure 10 highlights "interesting patterns" specifically in OLMo-7B. Despite this, the crucial experiment in Table 2, which aims to interpret these associations by checking correlations with similarity measures, was conducted on OLMo-1B. This choice is questionable, as an analysis on the more complex OLMo-7B would have been more compelling and could have yielded different insights into the nature of forgetting in more capable models.
2. The paper concludes from Table 2 that similarity measures "correlate poorly" with forgetting, dismissing them as weak predictors. However, this conclusion may be strong. For instance, the positive correlation for ⟨Gradient, Gradient⟩, while small, suggests that alignment in gradient space does contribute to forgetting.
3. The primary metrics, log perplexity and Token F1, show statistically significant but numerically small improvements (e.g., in Figure 5). This raises questions about the practical impact of the mitigation. A more direct and functional evaluation of forgetting would be to measure performance degradation on a specific, previously fine-tuned task (Task A) after the model learns subsequent new tasks (B, C, etc.). By constructing an association matrix between Task A's examples and the new downstream tasks, one could directly quantify the retention of a specific capability, which is a more intuitive and practical measure of forgetting than perplexity changes across a vast pre-training corpus.
4. The entire method depends on access to upstream data for building the association matrix, making it unsuitable for closed-source models.
5. The best-performing offline mitigation variant doubles the fine-tuning computational cost by requiring a preliminary run, hindering its practical application.

---

> ### Author Rebuttal · Authors · 2025-07-30
>
> We thank the reviewer for the very thoughtful comments and questions.
>
> **W1: The crucial experiment in Table 2, which aims to interpret these associations by checking correlations with similarity measures, was conducted on OLMo-1B only**
>
> We thank the reviewer for the question and we agree that the results on OLMo-7B should be added to Table 2. We performed this new set of experiments and we will include the results in the final version of the paper.
>
> | Model                    | OLMo-1B        | OLMo-1B         | OLMo-7B        | OLMo-7B         |
> |-------------------------------|----------------|-----------------|----------------|-----------------|
> |  Metrics                            | Pearson $\rho$ | Spearman $\rho$ | Pearson $\rho$ | Spearman $\rho$ |
> | Textual (TF-IDF)              | -0.049         | -0.035          | 0.048          | 0.137           |
> | Textual (Representation)      | 0.021          | 0.017           | 0.059          | 0.062           |
> | <Gradient, Weight difference> | -0.003         | -0.009          | -0.004         | -0.004          |
> | <Gradient, Gradient>          | 0.061          | 0.052           | 0.013          | 0.002           |
> | OLMo-1B forgetting            | -              | -               | 0.395          | 0.453           |
>
>
> To summarize, on OLMo-7B, gradient-based similarity measures show similar or lower correlations with the actual forgetting compared to OLMo-1B.
>
> Nevertheless, we also notice that TF-IDF text similarity correlates positively with forgetting OLMo-7B but negatively in OLMo-1B. To better interpret this,  we measure how each component ($\alpha_k\beta_k^T$) in the association matrix $Z$ correlates with the TF-IDF text similarity. We highlight components where Spearman's |$\rho$| > 0.05
>
>
> |         | k=1   | k=2    | k=3   | k=4   | k=5    |
> |---------|-------|--------|-------|-------|--------|
> | OLMo-1B | **0.093** | **-0.075** | 0.010 | 0.001 | -0.003 |
> | OLMo-7B | 0.003 | **0.328**  | 0.030 | **0.080** | -0.020 |
>
> The results show that some components are significantly more positively or negatively correlated with the text similarity, and how the components (and their correlations to TF-IDF text similarity) differ between OLMo-1B and OLMo-7B models. We will update the paper with the new results and the analysis.
>
> **W2: The paper concludes that similarity measures "correlate poorly" with forgetting, dismissing them as weak predictors. However, this conclusion may be too strong.**
>
> We thank the reviewer for pointing out the issue. Yes, the correlations of similarity measures) to forgetting (especially TF-IDF and gradient inner products), despite small, is statistically significantly non-zero. Therefore, they can still be predictors of forgetting. We will improve the preciseness of the writing in the final version of the paper.
>
> **W3: A more direct and functional evaluation of forgetting would be to measure performance degradation on a specific, previously fine-tuned task (Task A) after the model learns subsequent new tasks (B, C, etc.). By constructing an association matrix between Task A's examples and the new downstream tasks, one could directly quantify the retention of a specific capability, which is a more intuitive and practical measure of forgetting than perplexity changes across a vast pre-training corpus.**
>
> We thank the reviewer for the thoughtful comment. Regarding evaluation on downstream tasks, the most relevant results presented in the paper should be forgetting of OLMo-7B-Instruct on the FLAN subset of Tulu, where we use the exact-match drop metrics (instead of perplexity). The association matrix is visualized in Figure 8 in Appendix, and we have forgetting prediction results (in prediction F1) and forgetting mitigation results (in Token F1 retention) included in Table 3 and Figure 5.
>
> In our new set of experiments, we measured forgetting on the LLM leaderboard task examples. We see the similar low-rank structures and will include the visualization and statistics in the final version of the paper. We agree with the reviewer that the part of functional evaluation of forgetting can be further extended. We will expand our discussions about this line of future work.
>
> **W4: The method depends on the access of the upstream data**
>
> We acknowledge this limitation in our Limitations section. In fact, we consider this limitation inherited from replay-based algorithms for mitigating forgetting. We still decided to study replay-based methods because they are shown to be scalable and effective when applicable in LLMs.
>
> **W5: The best-performing offline mitigation variant doubles the fine-tuning computational cost**
>
> We acknowledge the extra computational overhead of the offline variant. We advise future practitioners to apply the online variant proposed in the paper in cost sensitive scenarios.
>
> ---
>
> We thank the reviewer again for recognizing the strength of our paper and the thoughtful comments.

---

### Official Review · Reviewer_nRcp · 2025-07-03

**Clarity:** 3
**Significance:** 3
**Originality:** 3
**Rating:** 5
**Confidence:** 4

**Summary:**

The paper investigates forgetting in LLMs using a matrix factorization framework that captures interactions between training examples and downstream tasks. The central claim is that forgetting matrices can be approximated via low-rank decompositions, enabling prediction of forgetting effects prior to fine-tuning.

**Questions:**

1. What inductive biases do you think induce low-rank structure in forgetting matrices?
2. How stable is the low-rank approximation with respect to noisy or adversarial fine-tuning? Does PEFT (parameter efficient finetuning) such as LoRA change the conclusion?
3. Could this method be used *during* training to halt updates early when forgetting risk is high?

**Ethical Concerns:**

["NO or VERY MINOR ethics concerns only"]

**Final Justification:**

The authors have addressed my concerns regarding the two main weaknesses: (1) the lack of insights into why low-rankness holds and (2) the limited discussion on the definition of forgetting in language models. They have promised to incorporate relevant discussions into the revised version of the paper. Additionally, they plan to conduct an ablation study on the robustness analysis, specifically on noisy outputs or rejected responses in the FLAN subset of the UltraFeedback dataset, which will provide further context for the method and is appropriate for inclusion in the appendix. Accordingly, I have increased my score from 4 to 5.

**Limitations:**

discussed in the appendix

**Quality:**

2

**Strengths And Weaknesses:**

**Strengths:**
- Elegant abstraction of example-task interactions.
- Matrix completion to predict forgetting is novel, simple and straightforward.
- Strong empirical support on multiple architectures (MPT, Pythia, OLMo).

**Weaknesses:**
- Limited insight into *why* low-rank holds. is it architectural, training-dynamic, or data-driven? Do we have any theorectical conclusions about the low-rankness.
- The definition of forgetting is not formalized. It is unclear that if an example should be considered as forgotten when completely disappearing from any possible prompted generation or with no system prompts. Sometimes it might seem like the example is forgotten but when we twist the prompt slightly, we will see the model actually does not forget the example but just hide it from surface. For example, RLHF is believed to be useful for forgetting toxicity but recent research find that they might just hide the toxicity from the surface (Table 3, Chen et al 2024 Jet expansions of residual computation)


**Missing References:**

The paper can be improved by adding a general discussion of related work on forgetting. The paper focuses on the negative side of forgetting while recently the positive side of forgetting has also been researched. For example, "Improving Language Plasticity via Pretraining with Active Forgetting" (Chen et al., NeurIPS 2023), introduces a pretraining-time forgetting mechanism that proactively improves model plasticity. This is highly relevant to the stated goal of understanding and mitigating forgetting in language models but more from a positive angle. Including this work would help position your contribution more clearly within the broader literature that views forgetting not only as a side effect but also as a potentially useful objective. Additionally, the authors should consider citing "Digital Forgetting in Large Language Models: A Survey of Unlearning Methods" (Blanco-Justicia et al., 2024), which provides a comprehensive overview of the forgetting landscape, including both proactive and reactive forgetting for removing data points from LLMs.

Regarding the methodology for tracing what is forgotten, the paper currently relies on continuous perplexity shifts and exact match drops. These metrics are informative but may miss deeper semantic shifts. To complement this, the paper may benefit from referencing "Jet Expansions of Residual Computation" (Chen et al., 2024), which explores how structured knowledge flows through residual paths in LLMs. While not directly about forgetting, it provides a way to analyze internal changes between models before and after finetuning, effectively offering a model-level “diff” that may surface subtle patterns beyond output metrics. Although not explicitly about forgetting, these tools could enrich your analysis of how knowledge is forgotten during finetuning.

---

> ### Author Rebuttal · Authors · 2025-07-30
>
> We thank the reviewer for the thoughtful comments and the very helpful suggestions.
>
> **W1. Limited insight into why low-rank holds**
>
> We thank the reviewer for the thoughtful comment. As we have acknowledged in the Limitations section, we do not have a theoretical explanation of the low-rank structure at this moment.  We tried to connect it to first-order approximations of forgetting and low-rank subspaces where weight updates can reside, but the actual forgetting does not correlate strongly (Sec. 4.3). We leave theoretical and mechanistic interpretations of forgetting as future works.
>
> At this stage, our paper contributes to behavioral interpretability of the low-rank structure by studying how various factors affect the low-rank approximation (e.g. model size, knowledge coverage of LMs, etc.) with data-driven experiments.
>
> **Q1:  What inductive biases do you think induce low-rank structure in forgetting matrices?**
>
> We thank the reviewer for this insightful question. We expect that significant study is required to answer the question. We believe mechanistic interpretations on how knowledge is updated and encoded in models (e.g. the suggested reference Chen et al. 2024) is relevant. We will note future lines of work in the updated version of the paper.
>
> **Informal intuition about the low-rank structure**. one intuition of the low-rank structure is an analogy to human’s forgetting process. Compared to acquiring a new skill, forgetting depends less on the material (training data) that humans learn afterwards, resulting in the low-rank structure. We refer to the interference theory of the human forgetting process for this intuition.
>
> **W2: The definition of forgetting**
>
> Thank you for the thoughtful comment. We choose to examine forgetting with log perplexity increase or exact match drop because (1) they are connected to theoretical approximations of forgetting with first-order techniques (Sec. 3.3), allowing proper comparison and (2) we hoped to study forgetting on upstream examples where we are sure the models have seen before.
>
> We agree with the reviewer that forgetting defined in other ways can have different and interesting practical implications. This includes the "shallow forgetting" (i.e. forgetting that is recovered when twisting the prompts), and the positive side of the forgetting the reviewer suggested in the comment. We agree that including the discussions of these works can position our work paper and inspire future analysis of forgetting under these definitions. We thank the reviewer for the helpful references and we will discuss them in the related works section.
>
> **Q2-1: How stable is the low-rank approximation with respect to noisy or adversarial fine-tuning?**
>
> We thank the reviewer for the suggestion. In our new on-going experiments, we perform supervised fine-tuning over noisy outputs or rejected responses in the FLAN subset of the UltraFeedback dataset and will include the results in the final version of the paper.
>
> **Q2-2: Does PEFT (parameter efficient finetuning) such as LoRA change the conclusion?**
>
> Our empirical results presented in Appendix H and Table 9 shows LoRA improves low-rank approximation of the forgetting, indicating simpler associations between learned tasks and forgotten examples. Therefore, we do not expect LoRA to change the conclusion about the low-rank property.
>
> **Q3: Could this method be used during training to halt updates early when forgetting risk is high?**
>
> Thank you for bringing up this extension. Yes, we believe that by predicting forgetting during training allows us to efficiently determine early stopping of the training process. This should be computationally more efficient than evaluating actual forgetting over all upstream examples.
>
> ---
>
> We thank the reviewer again for recognizing the contribution of our paper and the helpful comments.

---

> > ### Comment · Reviewer_nRcp · 2025-08-06
> >
> > Thank you for the thoughtful rebuttal. I look forward to the final version of the paper. I have increased my score based on the authors' promise to incorporate the discussion on the definition of forgetting, to expand the limitations section to include future directions on exploring why low-rankness holds, potentially using theoretical or mechanistic interpretability methods, and to include the new results on adversarial finetuning.

---

### Official Review · Reviewer_J9oc · 2025-07-03

**Clarity:** 3
**Significance:** 1
**Originality:** 2
**Rating:** 3
**Confidence:** 3

**Summary:**

The dependency between forgotten upstream examples and the newly learned tasks can enable the mitigation of forgetting when fine-tuned. This work shows that the association matrices are often well-approximated by low-rank matrices, and matrix completion allows fast identification of the most forgotten examples.

**Questions:**

1. The experiments use only decoder-only models. Would this method also work effectively on encoder-only models or mixture-of-experts architectures? Extending the evaluation to diverse model architectures would help assess the generalizability of the approximation method.
2. Why is forgetting measured with log perplexity (weakness 3)? Why not use task-specific metrics, for example, context-aware accuracy or answer correctness for QA datasets? What justifies log perplexity as the primary metric? A more detailed explanation is needed.
3. Does the replay approach impact the learning of the new task? Replaying a mini-batch of upstream examples every few training steps could interfere with the training dynamics on the new task. Have you assessed this effect?
4. If the previous data is already known, why is matrix-based prediction necessary? One could directly measure actual forgetting using the original data, and those results could serve as ground truth for evaluating the proposed mitigation method. This seems circular… A clarification would be helpful.

**Ethical Concerns:**

["NO or VERY MINOR ethics concerns only"]

**Final Justification:**

As some of my concerns have been addressed, I have decided to raise my score to a 3.
- However, I remain unconvinced by the proposed mitigation strategy, which is presented as a key contribution. It seems not improving much compared to the random method.
- Additionally, the method relies on an idealized setting that assumes access to the original training data and the ability to replay the fine-tuning process after predicting the forget data. The assumptions may not hold in many practical scenarios.

**Limitations:**

yes

**Quality:**

1

**Strengths And Weaknesses:**

### Strengths

- Proposes a simple method that estimates the associations between learned tasks and forgotten examples through low-rank matrix approximation, and predicts forgotten examples by solving a matrix completion problem.
- Demonstrates a mitigation approach that leverages the predicted forgetting. The practical utility is validated by showing statistically significant reductions in forgetting.

### Weaknesses

- **The experimental setting is overly idealized.** A critical limitation is that the method assumes access to the previously trained data, which is often unavailable in practice due to expired licenses or unreleased datasets. This significantly limits the applicability of the method. In real-world scenarios, such as post-training an LLM on a downstream task, the previously used data is unknown, making it infeasible to predict forgotten samples.
- **The mitigation method’s performance is not consistently superior,** for example, in Figure 5(c), its performance is comparable to random sampling of upstream examples. This raises concerns about the accuracy of the predicted forgetting and the effectiveness of the proposed sampling strategy.
- **Log perplexity may not be an appropriate evaluation metric for assessing forgetting mitigation.** A more meaningful metric would capture the extent to which forgetting degrades model capability, such as benchmark-specific accuracy or task-relevant correctness (e.g., accuracy on math problems).

---

> ### Author Rebuttal · Authors · 2025-07-30
>
> We thank the reviewer for the thoughtful comments. We respond to the weaknesses and the questions suggested by the reviewer below.
>
> **W1: The experimental setting is overly idealized. A critical limitation is that the method assumes access to the previous training data, which is often unavailable in practice due to expired licenses or unreleased datasets.**
>
> Thank you for pointing out the limitation. Although we agree with the reviewer about the inapplicability of our approach under data unavailability (inherited from replay-based methods), we decided to study replay-based methods because they are shown to be scalable and effective when applicable, which probably explains why they are still actively studied for LLMs [1,2]. This setup applies to, for example, developers that own their model and the training data. We will better clarify the applicability scope of the proposed method following the reviewer’s comment. Currently, this limitation is discussed in the Limitations section in Appendix A.
>
>
> **W2: The mitigation method’s performance is not consistently superior, for example predict+replay performance is close to random replay in Figure 5(c)**
>
> We understand the concern from the reviewer regarding small numerical performance gaps like in Figure 5(c). At this stage, our main goal is to analyze patterns in forgetting and showcase the benefit of predicting forgetting. We leave more effective forgetting mitigation algorithms that utilize the predicted forgetting as future works.
>
> **Are the improvements consistent?** Yes - we measured statistical significance of the improvements with paired t-tests in all setups in Figure 5, and achieved p<0.05 or 0.01 in most of the setups. To elaborate, a small p-value in this test suggests over all 20 tasks, our method brings consistent (but probably small) improvements.
>
> **W3 / Q2: Question about using log perplexity over the upstream training data as an appropriate metric of forgetting. Why not use task-specific metrics, for example, context-aware accuracy or answer correctness for QA datasets? What justifies log perplexity as the primary metric?**
>
> We thank the reviewer for the thoughtful question. The reason why we chose log perplexity over upstream data as the main metrics is many-fold.
>
> **First, log perplexity bridges theoretical approximations and empirical observations of forgetting, allowing proper comparison (in Sec. 3.3).** Log perplexity, equivalent to the training loss of language modeling, is often approximated with first-order techniques (gradient x gradient, gradient x weight differences) in prior works [3,4], assuming small weight updates. These measures can be, in theory, taken as predictors of forgetting; which, however, do not correlate strongly with empirical forgetting under practical (longer) fine-tuning runs in our experiments (Table 2 in Sec 3.3.). To compare with such theoretical approximations, we decided to measure forgetting with log perplexity increase.
>
> **Second, we try to measure forgetting over examples that the model has surely been trained on, which is the upstream pre-training or instruction-tuning data.** We also dissect forgetting from confounders such as generalization effects on unseen tasks.
>
> We will significantly improve the clarity about the choice of log perplexity metrics in the final version of the paper.
>
> ----
>
> In addition, we understand the limitations of using log-perplexity as the only metrics and that performance drop measured on benchmark dataset is also very interesting.
>
> **We would like to highlight results in the paper that utilize metrics other than perplexity in the current version of the paper**. For instruction-tuned model OLMo-7B-Instruct, we measured binary exact-match drop on the FLAN subset of Tulu V2 as the metrics of forgetting. The dataset involves QA pairs of moderately diverse tasks, and the exact match metrics is close to the accuracy metrics the reviewer asked for. We visualized the association matrix (Figure 8) and analyzed its low-rank patterns (Figure 3(a) - OLMo-7B Inst F1). We measured the performance of predicting binary exact drop and showed statistically significant improvement of targeted replay in Figure 5(b). In summary, the results are very consistent with forgetting measured with log-perplexity.
>
> **Current results & future works about forgetting on benchmark datasets.** Currently, we showed and concluded that targeted replay does not improve LLM leaderboard tasks performance (Table 7, Appendix F). We leave more effective algorithms as future works. We will include example-level forgetting statistics on LLM leaderboard benchmark datasets and analyze their low-rank approximations in the final version of the paper.
>
> **Q1: Extension to MoE and other model architectures**
>
> We thank the reviewer for the suggestion. We have tried to make our empirical analysis as comprehensive as possible given computational resources available. We understand MoE models are getting increasingly important but at this stage we have to leave it as future works.
>
> **Q3: Does the replay approach impact the learning of the new task? Replaying a mini-batch of upstream examples every few training steps could interfere with the training dynamics on the new task.**
>
> We thank the reviewer for the insightful question. We have experiments included in Appendix that can address the reviewer’s question. Per the reviewer’s comment, we will move more discussions to the main body of the paper.
>
> In Appendix F and Table 6, we showed that targeted or random replay can achieve comparable or improved perplexity on the validation sets of the fine-tuned tasks. In some setups, the improvements are statistically significant, which we will annotate in the updated paper.
>
> In Appendix H and Table 10, we measured the correlations between forgetting under different replay strategies. We see replay can change the patterns of forgetting compared to no replay, but the change is mild.
>
>
> **Q4: If the previous data is already known, why is matrix-based prediction necessary? One could directly measure actual forgetting using the original data.**
>
> We thank the reviewer for the question. Although we assume access to the upstream data, it is  unknown how much the examples suffer from forgetting after fine-tuning on a new task. This can be expensive to acquire, as measuring forgetting requires running inference over the possibly massive upstream data. Our motivation of predicting forgetting with matrix completion instead of directly measuring them is to avoid this heavy computation, by leveraging the empirical low-rank property of the matrix. We compared computational costs of different replay strategies in Table 4. As the cost analysis suggests, we expect further increased benefit of predicting forgetting in larger-scale setups.
>
> An analogy we made in the paper is collaborative filtering in recommender systems, which saves cost by inferring the missing user-item interactions by leveraging the low-rank property of the data matrix, instead of directly querying the user.
>
> ----
>
> We thank the reviewer again for the thoughtful comments, and we will update the paper accordingly.
>
> > References
> >
> > [1] Ibrahim et al. Simple and Scalable Strategies to Continually Pre-train Large Language Models, 2024
> >
> > [2] Que et al. D-CPT Law: Domain-specific Continual Pre-Training Scaling Law for Large Language Models, NeurIPS’ 24
> >
> > [3] Lepez-Paz et al. Gradient episodic memory for continual learning, NeurIPS’ 17
> >
> > [4] Doan et al. A Theoretical Analysis of Catastrophic Forgetting through the NTK Overlap Matrix, AISTATS ‘21

---

> > ### Comment · Reviewer_J9oc · 2025-08-07
> >
> > Sorry for the late reply. Thank you for the detailed rebuttal. As some of my concerns have been addressed, I have decided to raise my score to a 3. However, I remain unconvinced by the proposed mitigation strategy, which is presented as a key contribution. Additionally, the method relies on an idealized setting that assumes access to the original training data and the ability to replay the fine-tuning process after predicting the forget data. The assumptions may not hold in many practical scenarios.

---

> ### Author Response · Authors · 2025-08-06
>
> Dear reviewer J9oc,
>
> We hope to hear back from you as the end of the author response period is coming close. We are happy to take further questions. If the questions are addressed, we will be happy to see the reviewer re-evaluate the quality and significance of this work.
>
> Thank you.

---

### Official Review · Reviewer_9wDJ · 2025-07-11

**Clarity:** 3
**Significance:** 3
**Originality:** 4
**Rating:** 6
**Confidence:** 5

**Summary:**

This paper explores how LLMs forget during finetuning. They do this by building forgetting matrices between “upstream” examples (i.e. examples that had good performance before finetuning) and new tasks (i.e. after finetuning). They propose a method to efficiently predict forgetting of upstream examples, as well as an algorithm to mitigate forgetting during LLM finetuning. This has many practical implications for LLM finetuning and dataset selection and will be of interest to a wide range of LLM practitioners.

**Questions:**

Q1: I assume all models except OLMo-7B-Instruct are base models

Q2: It would be helpful to include some explicit examples in the appendix of data pairs/tasks where forgetting is high and where forgetting is low. This would help the reader build some intuitions around the very dense matrix plots that are presented (I know the Appendix is already quite extensive)

Q3: Why is the matrix for OLMo-7B-Instruct much more different than the the matrices for the other models in Appendix Figure 6?

Q4: “We empirically notice that the associations between learned tasks and forgotten upstream examples are more complicated in more recent and capable LLMs” - is this simply because more of the finetuned datasets are more likely to be included in OLMo pretraining?

Q5: Why does OLMo2 have zero average forgetting in Figure 3(d)?

Q6: Why do you use all-distilroberta-v1 as your embedding model? This seems quite outdated

Q7: What exactly is the “ground truth forgetting” in Figure 5? Is it the performance of the model after full finetuning?

Q8: What are the pros and cons between online and offline MF in Table 4?

**Ethical Concerns:**

["NO or VERY MINOR ethics concerns only"]

**Final Justification:**

The reviewers have more than adequately addressed my concerns. I believe these results will be very useful for the ML community.

**Limitations:**

The authors have addressed the limitations and potential negative societal impact.

Minor issues:
* The citation for MPT and RedPajama is [Computer 2023] which seems incorrect. Seems that most papers cite the original MPT blogpost
* Line 129: “knwon” → known
* The schematic in Figure 4 is slightly confusing. It might be helpful to split into (a) and (b) sections
* Line 241: “The additive linear model approximate forgetting”

**Quality:**

4

**Strengths And Weaknesses:**

Strengths:
* Well written and well motivated. The authors find that associations between learned tasks and forgotten examples are often well approximated with low-rank matrices, and use this to motivate their main method: matrix completion over association matrices.
* This paper is a tour de force across models (OLMo, MPT, Pythia, OLMo2, OLMo-Instruct) and datasets (FLAN, Tulu, Dolly), with an extensive appendix

Weaknesses:
* This paper presents a lot of empirical results that are somewhat difficult to interpret (i.e. all the matrix plots in the appendix). Aside: This is very helpful/useful information, and I wonder if the paper would benefit from some kind of interactive website that allows users to explore the results with more granularity.
* The final results in Figure 5 are quite compelling. However as a reader I am left wondering how much more efficient this result is in practice. Is this method and approach something we should all be using, or just something scientifically interesting?

---

> ### Author Rebuttal · Authors · 2025-07-30
>
> We thank the reviewer for the very thoughtful comments and questions.
>
> **W1/Q2: Some empirical results are hard to interpret (e.g. the matrix plot in the Appendix). Interactive websites to allow users to explore the results with more granularity can be helpful; Explicit examples in the appendix of data pairs/tasks where forgetting is high and where forgetting is low can be included.**
>
> We thank the reviewer for the suggestions. We have already started building an interactive online explorer of the matrix that also shows contents of the data pairs. We will improve the details and clarity of the figures and captions in the final version of the paper.
>
> **W2: How much more efficient the use of predict-then-replay result is in practice? Is this method and approach something we should all be using, or just something scientifically interesting?**
>
> Thank you for the thoughtful question. We consider that the predict-then-replay approach can potentially be extended to practical setups, and that the computational cost benefit can signify. As our cost analysis in Sec. 4.3 and Table 6 shows, the computational cost benefit of predict-then-replay increases when the upstream data is large, and when the same model is repetitively fine-tuned for different tasks.
>
> **Q1: I assume all models except OLMo-7B-Instruct are base models**
>
> Yes, we referred to non-instruction-tuned models as base models. We will improve the clarity in the writing.
>
> **Q3: Why is the matrix for OLMo-7B-Instruct much more different than the matrices for the other models in Appendix Figure 6?**
>
> We thank the reviewer for noticing this pattern. We believe it is mainly attributed to the differences in the data used in OLMo-7B-Instruct experiments.  To elaborate, we evaluated forgetting over the instruction-tuning data, rather than the pretraining corpora in other non-instruction-tuned model experiments. As there are increased chances of task or format similarity between the fine-tuned tasks and the instruction-tuning data (compared to pretraining data), forgetting (log perplexity increase) is sometimes negative, shown in blue pixels. Besides, the set of fine-tuning tasks (T_1..N, y-axis labels of Figure 6) are also different from non-instruction-tuned models, as we make sure the fine-tuning tasks are unseen by the models in the instruction-tuning phase.
>
> **Q4: “We empirically notice that the associations between learned tasks and forgotten upstream examples are more complicated in more recent and capable LLMs” - is this simply because more of the finetuned datasets are more likely to be included in OLMo pretraining?**
>
> We thank the reviewer for this insightful question. For open-data LLMs like OLMo and OLMo2, the source of the upstream data is released and splitted into dedicated domains. As far as we know, the fine-tuning datasets are not (intentionally) included in the pretraining phase (although a small portion may slip into the commoncrawl partition of the corpora). Therefore, we do not consider that the complexity of the pattern can be solely attributed to the training data inclusion.
>
> Besides, although our Appendix J suggests that inclusion of exact examples of the learned task in the upstream examples can lead to more complicated patterns of forgetting, we further notice that including the data from the same distribution (instead of the exact examples) results in the same patterns of forgetting. Therefore, we hypothesize that complexity depends on the overlap in the “distribution” or “knowledge” level, instead of the exact examples. We consider that understanding what attributes to the complexity of the patterns can require extensive study, which we leave as the future work.
>
> **Q5: Why does OLMo2 have zero forgetting in Figure 3(d)?**
>
> Thank you for noticing this pattern. Although forgetting appears on the zero line, the forgetting is actually non zero. We present additional statistics below.
>
> | Model                  | OLMo2-7B | OLMo-7B |
> |------------------------|----------|---------|
> | Avg. Forgetting        | 0.0037   | 0.0810  |
> | Std. Forgetting        | 0.0076   | 0.0646  |
> | Percentage where forgetting > 0 | 81.76%    | 96.23%   |
>
> We will include these statistics and the visualization of the association matrix of OLMo2-7B in the final version. We further note the low forgetting does not mean the OLMo2 has lost plasticity for updates. This is supported by the drop of perplexity on the fine-tuned tasks in a level similar to OLMo models.
>
> Nevertheless, we notice that OLMo2 models are much more resistant to forgetting when learning the new tasks. This can depend on many factors, such as flat loss minima (so that the upstream data perplexity is resistant to any updates in weights, often affected by choice of optimizers, regularization, dropout etc. during pre-training) [1], and whether the new task data is considered significantly out-of-distribution by the model (so that learning new task updates model parameters in a direction that causes forgetting). There are also empirical studies showing how forgetting depends on pre-trained models such as [2,3].
>
> > [1] Mirzadeh et al. Understanding the Role of Training Regimes in Continual Learning, NeurIPS 2021
> >
> > [2] Scialom et al. Fine-tuned Language Models are Continual Learners, EMNLP 2022
> >
> > [3] Luo et al. An Empirical Study of Catastrophic Forgetting in Large Language Models During Continual Fine-tuning
>
> **Q6: Why do you use all-distilroberta-v1 as your embedding model? This seems quite outdated**
>
> We thank the reviewer for the suggestion. We will update Table 2 and Table 3 with the results of OpenAI text embedding models in the final version of the paper.
>
> **Q7: What exactly is the “ground truth forgetting” in Figure 5?**
>
> We are sorry about the clarity issue. The dash line “ground truth forgetting” here means the replay examples are upweighted according to the actual forgetting, instead of the predicted forgetting.
>
> This result is included as a reference for predict-then-replaying approaches; computing the actual forgetting is expensive in practice, as it requires running LLM inference over the entire upstream data.
>
> **Q8: What are the pros and cons between online and offline MF?**
>
> The advantage of online MF is computation efficiency. As summarized in Table 4, online MF does not incur extra fine-tuning. However, the prediction of forgetting is less accurate as it is based on forgetting under a fraction of total fine-tuning steps. This is also evidenced by the subpar forgetting mitigation performance compared to the offline variant in Figure 5 in setups such as OLMo-1B.
>
> **Minor writing and presentation issues**
>
> We thank the reviewer for spotting the issues and the suggestions to enhance the figure presentation. We will update the paper accordingly.
>
> ---
>
> We thank the reviewer again for recognizing the contribution of our paper and the insightful questions.

---

### Decision · Program_Chairs · 2025-09-17

**Decision:**

Accept (poster)

**Comment:**

This paper investigates LLM forgetting by showing that the associations between newly learned tasks and forgotten upstream examples form a low-rank matrix. Building on this observation, the authors propose a method to efficiently predict forgetting of upstream examples, as well as an algorithm to mitigate forgetting during LLM finetuning. The approach is supported by strong empirical results across multiple models (OLMo, MPT, Pythia, OLMo2, OLMo-Instruct) and datasets (FLAN, Tulu, Dolly). Most of the reviewers’ concerns have been adequately addressed. Remaining issues include the marginal improvement of the proposed method and the practicality of assuming availability of the original training data.